# Estimation of the domestic water demand– supply scenario and its key driving factors in the Islamabad-Rawalpindi Metropolitan Area, Pakistan

Sidra Aman Rana[1], Syeda Maria Ali[1]*, Muhammad Ashraf[2] Naveed Iqbal Gondal[2]
Sadia Rahman[1], Nadia Akhtar[1]

1 Department of Environmental Science, Faculty of Sciences (FoS), International Islamic University Islamabad (IIUI), Islamabad, Pakistan, 2 Pakistan Council of Research in Water Resources (PCRWR), Islamabad, Pakistan

* maria.ali@iiu.edu.pk

## Abstract

Domestic water availability is intricately linked with a growing population, urbanization, and associated land use/land cover change (LULCC). Precise and periodic assessments of water supply and demand are imperative for the sustainability of urban ecosystems. The present study provides a situational analysis of the domestic water supply and demand and LULC to quantify their impacts on water resources in the rapidly growing water scarce metropolitan area of Islamabad-Rawalpindi. Data regarding demography, water demand and supply were collected from concerned government departments, and the water demand-supply balance was projected for the years 2021–2050 through simple equations. Two remotely sensed satellite imageries (Landsat 8 OLI and Landsat 5 TM) were retrieved and supervised classification was performed for land use land cover (LULC) analysis during last two decades 1999 and 2019. The results revealed that the current domestic water supply is 65 million gallons per day (MGD) and 54.074 MGD in Islamabad and Rawalpindi, respectively. The per capita water availability is estimated as 60 GPCD (gallons per capita per day) in Islamabad, whereas it is 76.40 GPCD in Rawalpindi. Water demand has outstripped conventional supplies, leading to deficits of 52 MGD and 18.50 MGD in Islamabad and Rawalpindi, respectively. The groundwater depth has decreased from 22.8 m to 56 m in Islamabad and from 37.8 m to 59 m in Rawalpindi. This decline is positively correlated with the density of installed tube wells, borehole wells, demographic factors and LULCC. LULC analysis depicts rapid decline in open land which has been replaced by settlements. Moreover, the area of open land decreased significantly from 68.46% to 18.92%whereas the area under the built environment increased significantly from 7.77% to 36.51% from 1999—2019. However, increase in the forest cover and water was insignificant. This land transformation contributed to a reduction in the water table depth and recharge to aquifers and escalating water demand. The study concluded that the groundwater resources of twin cities are under stress. In addition, forecasted water demand will also increase in the future with increasing population, economic growth and lifestyle changes while increasing groundwater abstraction, and diminished water

**Data availability statement:** All relevant data are within the paper and its Supporting information files.

**Funding:** The authors received no specific funding for this work.

**Competing interests:** The authors have declared no competing interests.

infiltration rates due to urban expansion will compromise water availability for future generations. The findings provide valuable information to concerned agencies, policy makers and stakeholders to take appropriate long-term measures to address repercussions of unplanned urbanization and LULC to ensure a sustainable water supply to the urban populace.

## Introduction

Freshwater supplies are gradually diminishing in urban areas across the globe due to rapid population growth, urbanization, changes in land use land cover (LULC) and climate [1]. Unrestricted urban growth has placed tremendous pressure on vulnerable hydrological systems, particularly in areas with high population density. LULC depicts natural and cultural features found across earth landscapes, such as vegetation cover, water bodies and human structures, and is currently considered major human interference. Flash flooding, lowering water table and declining recharge induced by LULC changes increase with increasing water demand, and deteriorating water quality further stresses surface and groundwater resources in already water scarce regions [2]. Globally, land use conversions can contribute directly in run-off increase to about 0.08mm/year. In addition, per capita water demand, which has almost doubled compared with the rate of population growth since the last century, points toward the additional water requirements induced by lifestyle changes. Numerous studies have demonstrated that a positive correlation exists between economic growth and water demand as a result of increases in the agricultural, energy, industrial and residential sectors. The situation is likely to exacerbate in the coming years with a growing population, rapid urbanization and climate change.

Consequently, countries are facing challenges in maintaining adequate and safe urban water supplies to ensure water security [3]. Water security is a "central theme" for sustainable development goals 6 (SDG-6) and SDG-14 (life below water), while it has strong linkages with other SDGs, including SDG-2 (zero hunger), SDG-7 (affordable and clean energy), SDG-13 (life on land), SDG-3 (health and wellbeing), SDG-11 (sustainable cities and communities) and SDG-12 (responsible consumption and production) [2]. Clean and sustained water supplies are needed not only for meeting domestic but also for agricultural and industrial demand. Adequate access to safe and reliable water supplies contributes significantly to improved health and increased overall productivity [3].

According to the World Health Organization (WHO), approximately 75 liters of water per capita per day are needed to fulfill household demands, of which 50 liters per capita per day are needed to maintain basic health and hygiene. However, the lowest water requirement to circumvent food and health implications is 1,000 cubic meters per capita per year [3,4]. Applying these criteria, nearly 1.6 billion people (constituting one quarter of the population) are estimated to be currently experiencing water scarcity. The world population will increase to 10 billion in 2050, resulting in a 55% increase in water demand, including water for food and energy. In this context, Asia and the Pacific region are more vulnerable, as they are expected to host more than 60% of its population dwelling in urban areas by the year 2050 [5,6], which will further strain the already stressed water supply system. The availability of adequate quantities of water is the basic requirement for most socioeconomic development in various sectors, such as agriculture, industry, aquaculture, navigation, and power generation. Contextually, a high-quality water supply system should serve two primary functions, "adequacy" and "reliability", by ensuring three basic elements, i.e., water availability, water quality and delivery [7].

The factors that influence water demand and supply are physical and economic. The physical availability of water is a function of the freshwater sources present in an area and innovative techniques to augment supplies such as rainwater harvesting and desalination. Conversely, economic scarcity is the ability of the population to access safe water resources. Like physical availability, economic availability also depends on the socioeconomic resilience of the government and the technological efficiency of the water supply system. Traditionally, the provision of basic environmental services such as water supply and sanitation is the responsibility of the local government. Government institutions have low institutional capacity, particularly in developing states such as Pakistan, where the water supply infrastructure and services are extremely poor, outdated and inadequate. Moreover, financial constraints limit the construction and maintenance of new infrastructure to improve urban water supplies [8].

Pakistan is one of the most populous countries in the world, with a total population of 242.8– million in January 2024, which is expected to reach 250 million in 2025 and 335 million by the year 2050 [9]. With the highest rate of urbanization among South Asian countries, the urban population in Pakistan is expected to increase further (52%) by the year 2025. Specifically, water demand is escalating at a rate of 10% annually in Pakistan and is expected to increase further in the metropolitan region [10]. Groundwater withdrawal increased approximately six folds from 1965-2020 in the country [11]. An estimation of total water demand in Pakistan revealed that the agricultural sector is the largest consumer (95%) of water, followed by the domestic sector (2–3%) and the industrial sector (2%) [12]. However, studies have failed to present the water required to maintain essential ecological functions and environmental flow in water bodies. The United Nation Food and Agricultural Organization (UNFAO) estimates of 2018 highlighted that Pakistan's freshwater resources are under tremendous stress, as reflected by the high pressure value of 74% and dependency ratio of 78. Pakistan is also among the 4th largest abstractors of groundwater in the world; presently, 62 km³ is exploited through a network of public and private tube wells.

The National Drinking Water Policy (2009) defines safe access to water resources in Pakistan as the daily availability of 120 liters per person in urban areas and 5 liters per person in rural areas. However, recent data indicate that per capita water availability sharply declined from 5600 cubic meters per capita per year in 1951 to less than 1000 cubic meters per capita in 2018, with a decrease in annual water availability to approximately 860 cubic meters per capita. If timely mitigation measures are not implemented, the same trend will likely continue, resulting in a decrease in per capita availability to 700 m³ by 2025 [13]. On the other hand, it is anticipated that domestic water demand for drinking purposes will experience manifold increases from the present 7.56 MAF to 32.49 MAF in 2050 and 10.37 MAF in 2025 [12]. Ostensibly, water scarcity in Pakistan is the outcome of water demand and supply imbalances and is considered a prime stumbling block in achieving the SDGs. Water scarcity has negative implications for the socioeconomic development of a region, as it threatens water, food, energy security, the household economy, manufacturing, transportation, and trade flows and consequently causes distress, suffering, chaos, and poverty. The vicious cycle is likely exacerbated with increasing pumping, production and treatment costs of water. Moreover, inadequate and polluted water supplies have become the leading cause of various diseases and mortalities in low- and middle-income countries such as Pakistan [14,15]. This situation calls for adequate urban water demand management on a warning basis to address present and future water demand–supply imbalances and their key driving factors [7](1). Several studies, such as North Carolina and South Carolina [1]; Dhaka, Bangladesh [6]; Addis Ababa city [9]; Dilla Town Ethiopia [3]; Madurai City, India [4] Hyderabad, India [2] (; and Lahore [16] and Peshawar [14], have been conducted in rapidly urbanized metropolises to address the urban water supply–demand imbalances occurred due to rapid population growth, unplanned

urbanization, socioeconomic development, change in life style, climatic changes, poor infra-structure and water losses for improving the long-term sustainability of urban water resources. Like all these metropolises, the water availability situation in the study area is quite distressing. Inhabitants of twin cities depend on groundwater, which is alarmingly depleted due to LULC transformation, unplanned urbanization, industrial progress, rapid population growth and high abstraction rates [16]. In recent years, a sharp decline in groundwater levels has been observed within the study area [17]. Water shortages have become a major predicament in the development of twin cities. In this scenario, the formulation and implementation of efficient water management strategies has become imperative to guarantee a sustained water supply in twin cities. Water managers, practitioners, policy makers and decision makers necessitate scientific, evidence-based and economically feasible solutions to surmount urban water demand challenges. The literature has revealed that comprehensive studies pertaining to estimations of domestic water demand and supply scenarios (current and forecasted) in the context of demographic trends, LULC changes and their impacts on water resources in twin cities are not available thus far. Numerous hydrological studies conducted in the twin metropolis are addressing water quality issues in source and filtered water to ensure public health.Therefore, basic aim of this study was to assess the impacts of LULC and demographic factors on water resources particularly current and future domestic water demand-supply scenario in twin metropolis of Islamabad-Rawalpindi to fill this knowledge gap. The main objectives of the present study were: 1) to assess current and projected water demand and supply scenarios in twin cities and 2) to relate urban demographic trends as well as the density of tube wells with water table decline in twin cities. 3). to relate land use land cover change (LULCC) to water demand and supply and water table decline in twin cities. This concrete baseline information will provide a roadmap to concerned policy makers, water planners and local civic agencies such as the Capital Development Authority (CDA), the Rawalpindi Water and Sanitation Agency (WASA) and the Rawalpindi Cantonment Broad (RCB) to devise and implement appropriate strategies to sustainably overcome water demand and supply challenges in twin cities.

## Materials and methods

### Study area description

The present study was carried out in the federal capital of Pakistan, Islamabad, and the adjacent city of Rawalpindi, which are commonly called twin cities. Islamabad is located at the base of Margallah Hills, while Rawalpindi is located at the northern fringe of the Potohar Plateau, Punjab. The Islamabad-Rawalpindi metropolitan stretches from 33°30′ and 33°50′ N (latitudes) and 72°45′ and 73°30′ E (longitudes), covering areas of 906.50km$^2$ and 259 km$^2$, respectively (Fig 1).

Climatically, the area falls in a humid to semiarid zone with moderate summers and winters. The rainfall pattern is erratic, and the annual precipitation in the study area ranges from 990 mm to 1,000 mm [17–19]. The maximum precipitation (79%) occurs from June to September and ranges up to 1750 mm due to the monsoon system, while the western depression also experiences sufficient downpour during the winter season to support the ecosystem [19]. The topography is rugged, varies in elevation and comprises mostly steep slopes and gullies [20]. The population in twin cities has increased rapidly due to rapid rural–urban migration influx [19]. According to the latest National Census 2017, Islamabad is the 9th largest city with a population of 1.5 million, and Rawalpindi is the 4th largest city with a population of 2.2 million. Furthermore, this metropolitan area is of particular economic concern, contributing 2.86 USD and 3.96 USD, respectively, to the country's gross domestic product (GDP) [21].

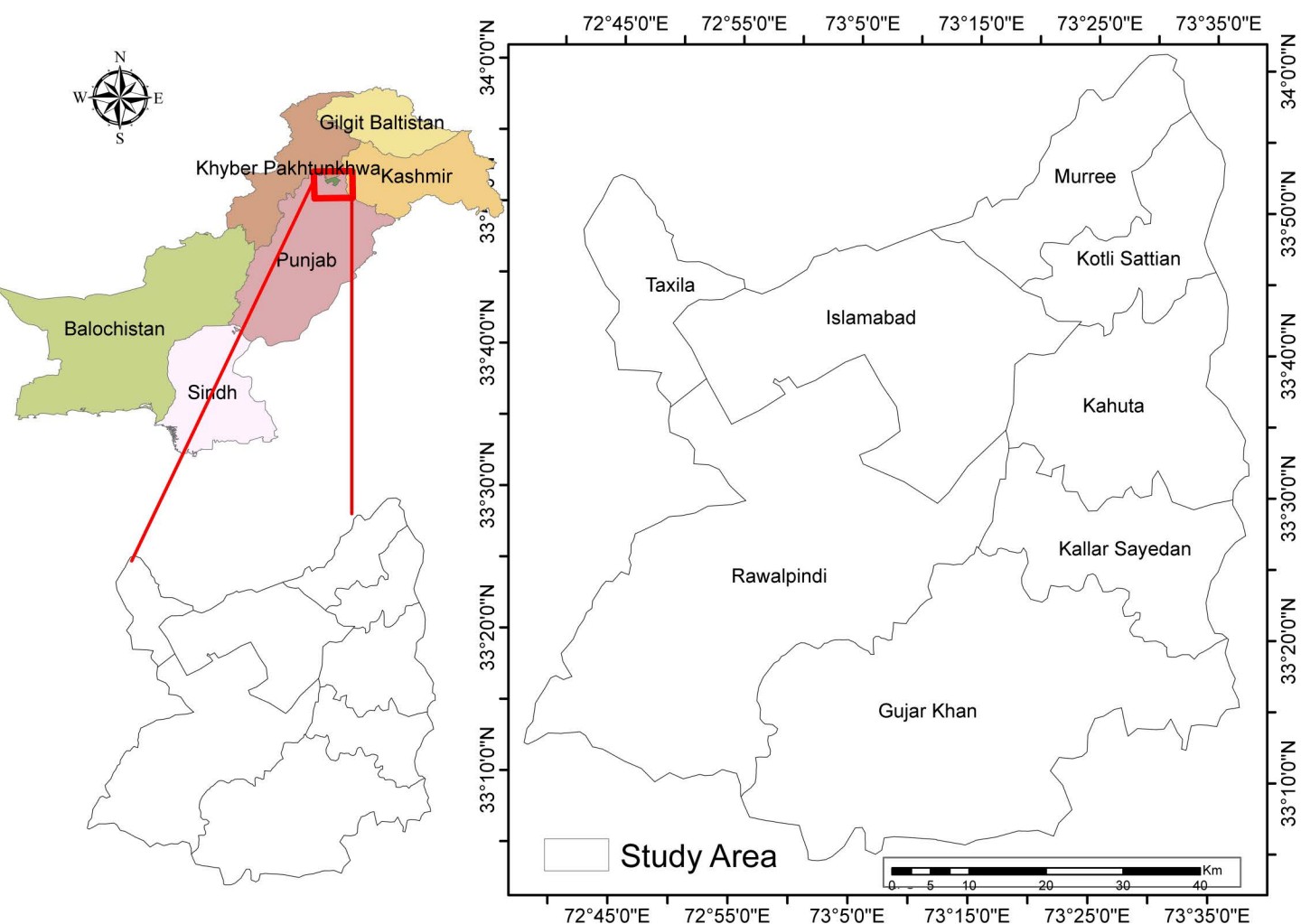

**Fig 1. Location map of the Islamabad and Rawalpindi metropolitan areas.**

The water quantity in most parts of twin cities is jeopardized because of over pumping, land use changes and urban, industrial and agricultural runoff [22]. On the other hand, per capita water demand is constantly increasing in twin cities in response to rapid population growth, lifestyle changes and economic development [23].

### Data collection

Fig 2 portrays the methodological framework adopted for this study. The current study relied on primary and secondary data acquired from various organizations through extensive interviews and field and literature surveys to estimate current and future water demand and supply. A semi structured questionnaire was designed for interviews to collect recent information regarding water supply and demand from officials of the water supply departments of CDA, WASA and RCB of both cities (See S1 File). Key informants (N = 5) were selected from the water distribution departments of these authorities. Secondary data regarding past trends in water supply and demand were obtained from official reports and websites of organizations responsible for the management and distribution of water in twin cities. In addition, information about water demand, water supply, existing tube wells, location, feeding area,

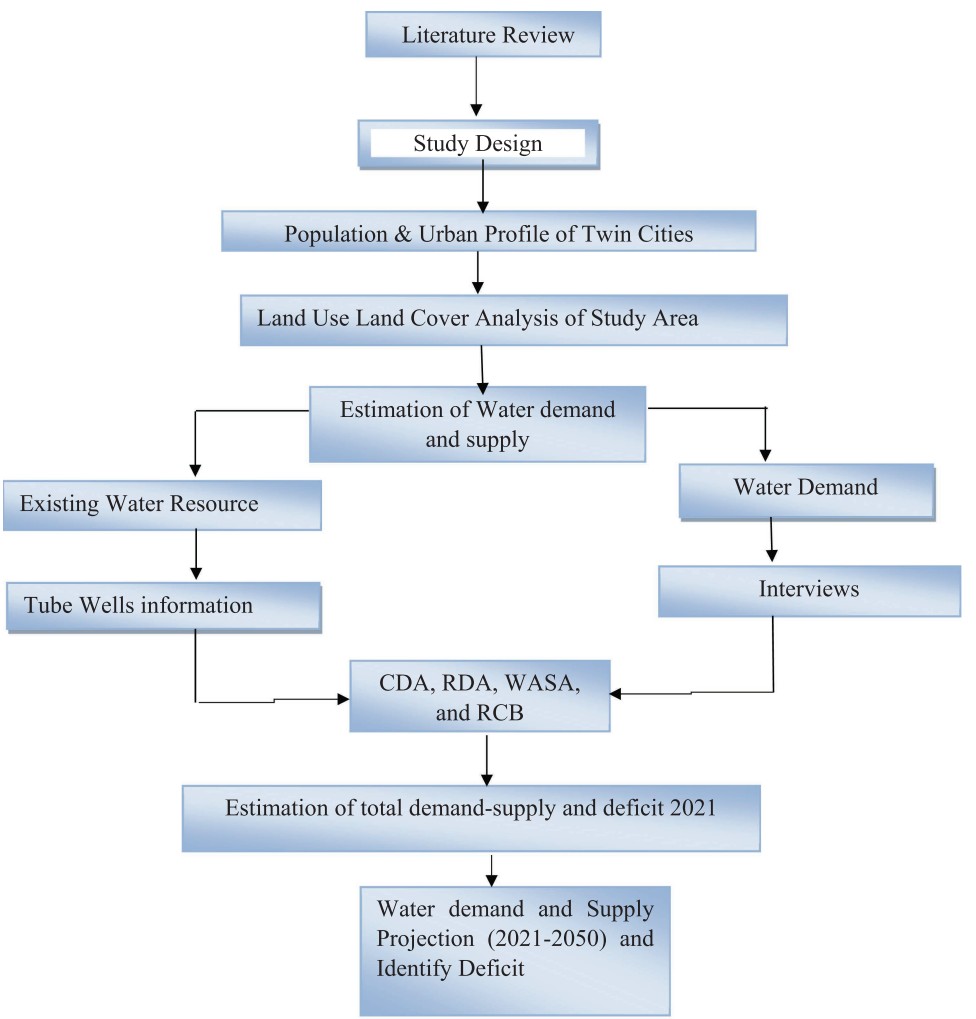

**Fig 2. Methodology flow chart.**

groundwater level and per capita water demand in Islamabad and Rawalpindi city was also collected from the CDA, RDA, WASA and RCB. Demographic trends (population size, population growth, population density, urban growth rate, housing units, and migration rates) for the period of 1951–2017 in twin cities were acquired from the Pakistan Bureau of Statistics (PBS) and the National Institute of Population Studies (NIPS). All primary and secondary data were collected from 2020–2021 during the COVID-19 pandemic. The secondary information collected from concerned civic agencies were cross verified through authentic sources to ensure accuracy and validity of results of present study. Present study was conducted in collaboration with PCRWR, key water experts and stakeholders. These water experts (N = 8) were consulted during research design, data collection, analysis, forecasting, interpretation as well as devising sustainable problem based solutions for validity and robustness of present study outcomes. The details of these experts were given in supplementary information.

## Total water supplied

Data regarding total water supplied from surface and ground water sources (tube wells and private bore wells) were collected from the water supply directorates of CDA, WASA and RCB.

## Total domestic water demand

On the basis of the per capita water consumption data obtained through CDA, RDA and WASA, total domestic water demand was calculated via the following formula [24] UN 2005 (1):

$$Q = K_1(K_d q P) \tag{1}$$

where Q is the domestic water demand, $K_1$ is the distribution water loss, $k_d$ is the mean daily water consumption per household, q is the mean per capita water consumption, and P is the total number of inhabitants. For the projection of water demand and estimation of future water availability, the population growth equation was used (2). The geometric growth model is presented as follows:

$$P_{t+n} = P_t \left( I + r \right) n \tag{2}$$

where $P_{t+n}$ = population at time, $P_t$ = population at present, r = rate of population growth per unit time, and n = length of time for which the projection is made.

Data regarding distribution water losses ($K_1$), mean daily water consumption per household ($K_d$), and mean per capita water consumption (q) were collected from officials and reports of water distribution authorities, i.e., CDA (for Islamabad), WASA and RCB (for Rawalpindi). These authorities supply water to more than 80% of the population residing in twin cities. The peak water demand was calculated by multiplying the average domestic demand by a factor of 1.8 [25].

## Water supply–demand gap

The water supply–demand gap was calculated by subtracting the total domestic water demand from the water supplied to the city dwellers.

## Forecasting domestic water supply and demand scenario for twin cities from 2025–2050

Data were processed to estimate present and future water demand and supply in Islamabad-Rawalpindi. The year 2017 was considered the base year for analyzing water demand, as the latest population census data were available only for this year. Subsequent projections were made for the years 2025 and 2050 to forecast the water demand and supply for the three different scenarios.

Scenario A was taken as the business-as-usual scenario considering the lack of improvement in water supply networks. Moreover, this scenario was based on the assumption that urban water demand will continue to increase with the same trend in the coming years, considering 35% water losses during distribution.

Scenario B was based on the assumption that significant improvement in the water supply infrastructure would result in a reduction in water loss of up to 10% (minimum criteria to be met in the case of developing nations).

Scenario C considers the installation of a highly efficient water supply system in the future to bridge the water supply-demand gap.

For this purpose, the demographic trends and water demand and supply forecasting scenarios were analyzed via Microsoft Excel and presented via graphs. This method is widely applicable for growing towns and cities with a vast scope of urban expansion [7–10]. The results are provided in the Supplementary Material as S6 and S7 Tables.

## LULC change and groundwater decline

**Data acquisition.** For the present study, satellite data were acquired mainly from three different sources. High resolution remote sensing imagery Satellite data was downloaded for free from earthexplorer.usgs.gov, www.glovis.usgs.gov and multitemporal Landsat satellite data from 1999 to 2019 for the LULC analysis of the study area. The satellite data for Landsat 5 Thematic mapper, Landsat 7 and the Landsat 8 operational land imager (OLI) with 30 m resolution. The paths/rows followed for L5TM, L7 ETM + and L8 OLI-TIRS were 150/36 and 150/37, respectively to cover entire study area.

**Data analysis through GIS.** GIS analysis was performed in a series of steps involving the preparation and processing of data with the help of different software, viz. ArcMap 10.1, ERDAS Imagine 2014, ENVI 5.3, Google Earth and Microsoft Excel. Data preparation included preprocessing of satellite data in ERDAS Imagine 2014. The satellite images were subjected to atmospheric and geometric corrections for the removal of cloud cover, image enhancement and conversion of DN values into reflectances. As the Landsat image has a variety of raster bands, stacking was performed to produce a single raster composite for further processing. After stacking, the next step involved mosaicking both tiles, and finally, sub-setting was performed to extract the area of interest to reduce the spatial resolution. The thermal band of Landsat was also preprocessed, and its resolution was resampled from 120 meters to 30 meters.

**Supervised classification for LULC.** Recognition of entities and phenomena on earth is the main focus of remote sensing, and classification is the best-fit technique for this purpose [26]. After preprocessing, supervised classification was applied to the satellite image. Supervised classification is a quantitative process in which each pixel is allocated to an individual class [27–29]. Four LULC classes were formed, namely, vegetation, open land, built-up areas and water features. The descriptions of these classes are given in Table 1.

The initial and essential step of supervised classification is the collection of training samples for each class into a signature file. After defining the LULC categories, 200–300 training samples were collected for each class. The objects were identified on the basis of their spectral signatures in different bands. Every object on Earth reflects and absorbs solar energy and has typical behavior represented in each wavelength of the solar spectrum, known as the spectral profile of the object. For example, water reflects blue wavelengths but absorbs infrared energy, thus creating a profile of reflectance and absorbance in different bands of the raster. The objects were placed in a specific class with the best match, and then the classification process was executed. The maximum likelihood algorithm was used to generate classified images of land use and land cover in the study area. The maximum likelihood algorithm places the pixels in the class with the highest probability [30]. It is the most efficient and widely used method thus far and has been proven to yield the best results for analyzing land dynamics. The two images were classified via a similar method to derive the final LULC change for the past 20 years. Finally, the area was calculated for each land cover class.

**Accuracy analysis.** Ground truthing was used to classify the dubious areas by using data from Google Earth and existing maps of the study area.

**Table 1. Major landuse land cover (LULC) classes in the Islamabad-Rawalpindi from 1999 to 2019.**

| Sr. no | Class name | Description |
|---|---|---|
| 1 | Vegetation | Includes the forest cover, agricultural land, vegetation in parks and green belt |
| 2 | Open land | Includes the barren land, rocks, decertified land and fallow land. |
| 3 | Built up | Commercial buildings, residential blocks, roads, bridges and other infra structure. |
| 4 | Water | Rivers, lakes, dams, small streams and nullahs. |

## Results and discussion

### A situational analysis of water resources in twin cities

**Groundwater resources.** The findings suggest that groundwater constitutes an indispensable source of water in the domestic, irrigation and industrial sectors in twin cities. The groundwater resources in twin cities are primarily stored and discharged from recent Quaternary alluvial deposits. The aquifers in the study area are superimposed aquifer systems such as superficial, deep and scattered aquifers and are therefore not suitable for large-scale production. Maqsoom et al. (2022) reported that only 15% of the region is the best zone for groundwater extraction [31]. The documented total groundwater availability is approximately 86 MGD [32]. Groundwater recharge occurs primarily from precipitation (80–90%) and partly from natural streams, drains, springs and rivers. The findings of the interviews corroborated the reported literature that, like most arid and semiarid areas of the world, more than 60% of the total water supply in Rawalpindi-Islamabad comes from the tube water/groundwater system [33]. Moreover, several private and municipal wells, hand pumps, and bore wells are also installed to meet the growing demand for freshwater [23]. However, accurate information about the total number of private tube wells and house bore wells operating in the study area is not currently available. A hydrogeological study reported by the National Engineering Services of Pakistan (NESPAK) revealed approximately 129 private tube wells in Islamabad [34].

The density of tube wells reflected a rapid growth trend with time, as 39 tube wells were installed in 1998, which grew to 80 and 278 in 2003 and 2007, respectively, in Rawalpindi city [17]. Currently, the number of tube wells has further increased to 421, 415 of which were functional from 2020–2021 in Rawalpindi. Similarly, a phenomenal increase in the number of tube wells is also observed in Islamabad city, whose number exceeds 205, of which 200 are functional, and the construction of several others is under process (Fig 3). The findings indicate that the total production from these functional tube wells was nearly 24.65 MGD in Islamabad and 38 MGD in Rawalpindi (Tables 2 and 3).

In addition, a substantial increase in the number of private tube wells installed by private, local and commercial homes (water tanker filling points) has notably altered the aquifer system [35]. Owing to the limited, intermittent and, in some cases, nonavailability of supply water in different areas of twin cities, dwellers make water supply arrangements by installing house bore wells [36]. Notably, Pakistan does not have any policies, acts or regulations to control groundwater abstraction. Consequently, housing societies and individuals are free to abstract groundwater from their properties. However, CDA has recently approved a policy for mandatory rainwater harvesting and recharge wells for residential and commercial buildings. In addition, the 50 recharge wells of the CDA replenished 10 million gallons of water (2022). Adopting these guidelines, the Federal Government Housing Authority successfully constructed groundwater recharge wells in the greenbelts of G-13 and F-13. However, similar initiatives at WASA and RCB are still lacking.

**Surface water resources.** The area has a natural gradient for draining surface water channels. Salient features of all the existing surface water resources found in twin metropolitan areas are given in Table 3 Nevertheless, as groundwater resources are limited, patchy and inconsistent, three major surface water reservoirs, namely, Rawal, Simly and Khanpur Lakes, serve as sources of water to the Rawalpindi-Islamabad metropolitan area. Hydrologically, the study area is drained by the Soan and Korang Rivers as well as numerous streams as their tributaries. The soan River emanating from Murree Hills is an important stream of the Potohar Region. A simple dam is constructed on this river. The Soan River discharges its content into the Indus River after covering a distance of approximately 250 km.

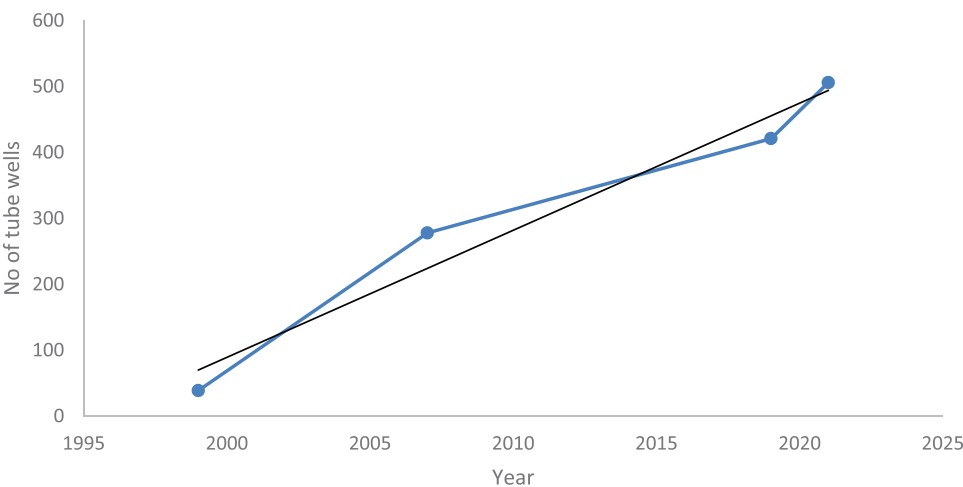

**Fig 3. Number of functional tube wells in twin cities.**

The Korang River is the major river originating from Margalla Hills. This river is dammed at Rawal Lake to address water supply shortages in twin cities [22]. Rawal Lake is located in southeastern Islamabad in isolated sections of Margallah Hills National Park [37]. This reservoir covers a surface area of approximately 8.8 km². The total storage capacity is 47,500 acre feet, whereas the live storage capacity is 43,000 acre feet. The raw lake caters to the water demands of inhabitants after filtration. The findings of the interviews revealed that the reservoir was supposed to provide 29 MGDs of water, which was reduced to 21 MGDs after the drought of 1972–73 and was still the same. A total of 19.5 MGD is supplied to Rawalpindi city, and 2.0 MGD is supplied to Islamabad [38].

A simple dam was constructed across the Soan River in 1983 approximately 30 km east of Islamabad Capital Territory (ICT). This dam is primarily recharged by spring water and snowmelt from Murree Hills. It served as the largest source of domestic water supply in Islamabad city. The mean annual water availability from the Simly Reservoir is reported as 68 MGD. As per the Water Supply Directorate of CDA, at present, the Simly dam supplies 26 MGDs to the federal capital.

The Khanpur dam is built across the Haro River in Haripur, Khyber Pakhtunkhwa (KPK), approximately 50 km away from Islamabad city. The total storage capacity of the Khanpur

**Table 2. Detailed summary of daily water production in Islamabad City.**

| Sr. No | Source | Design Capacity | Production Capacity | Actual Production (MGD) |
|---|---|---|---|---|
| 1. | Simly Dam | 42 | 39 | 20 |
| 2. | Khanpur Dam | 51 | 16.83 | 8.5 |
| 3. | Rawal Dam | 4 | 2 | 2 |
| 4. | Tube wells | 47 | 38 | 24.84 |
| 5. | Saidpur water works | 0.80 | 0.80 | 0.50 |
| 6. | Shahdara water works | 1.60 | 1.60 | 1.00 |
| 7. | Noorpur water works | 0.70 | 0.70 | – |
| 8. | Korang water works | 4.00 | 2.40 | – |
| 9. | Total | 149.1 | 101.33 | 56.84 |

Source: CDA, Production Division-I, dated: 02–06–2021

**Table 3. Salient features of three major surface water reservoirs for bulk water supply in the study area.**

| Salient Features | Khanpur Dam | Simly Dam | Rawal Dam |
|---|---|---|---|
| Location | Haripur | Islamabad | Islamabad Park Area |
| completion year | 1984-85 | 1982-83 | 1962 |
| Commence year | 1967 | 1972 | 1959 |
| Impoundment | Haro River | Soan River | Korang River |
| Purpose | Drinking –Irrigation-Industrial | Drinking -Irrigation | Drinking -Irrigation |
| Status | Operational | Operational | Operational |
| Geographical Location | 33.807°N 72.936°E | 33°43′08″N 73°20′25″E | 33°42′N 73°07′E |
| Type of dam | Earth and rock fill | Embankment/Earth fill | partly arched gravity dam (stone masonry) |
| Spillway type | Ogee type weir | Ogee overflow | Ogee gated structure |
| Surface Area | | 1.7 km² | 8.8 km² |
| Catchment area | 308 sq-miles | 153 km² | 275.2 km² |
| Height of dam | 167 feet | 80 m (262 ft) | 133.5 ft (40.7 m) |
| Over flow Depth | 35 feet | 20 feet | 19 feet |
| Length of dam | 1546 ft | 1,027 ft | 700 ft |
| Reduction | ------- | ------ | 27,000 acre feet |
| Live storage capacity | 91,500 acre-feet | 20,000 acre-feet | 43,000 acre-feet |
| Dead Storage capacity | 15,500 acre-feet | 8750 acre-feet | 4,500 acre-feet |
| Gross storage | 107,000 acre-feet | 28,750 acre-feet | 42,000 acre-feet |
| Controlling Division | WAPDA | Capital Development Authority (CDA) | Small Dams Organization (SDO) |
| Maximum flood level cusec | 182,000 | 90,684 | 120,000 |
| Canals | 2 | ---- | 2 |
| Left Bank Canal | Supplies water both for irrigation as well as municipal and industrial purposes. | ---- | Irrigation |
| Right Bank Canal | irrigation solely | ---- | Drinking water supply to Rawalpindi |
| Drinking water supply Capacity to Rawalpindi | RCB = 9 MGD; RWASA = 10 MGD | ---- | 22.00 |
| Drinking water supply Capacity Islamabad | 51 | 42 | 4 |

reservoir is 35463000 m³. The mean annual water availability is estimated at 198 MGD. This is a multipurpose dam that supplies water for irrigation and domestic and industrial use. Originally, the project was designed to supply 33 MGDs of water to Islamabad and 67 MGDs to Rawalpindi. However, interviews revealed that the Khanpur dam supplies 8.50 MGDs to the federal capital and 9 MGDs to Rawalpindi (of which 5 MGDs are received to RCB and 4MGDs to WASA) because of a reduction in the water level [39,40].

Another notable source of surface water is head work. Recently, four head works have been operating in Islamabad City. Korang River headwork, Saidpur headwork, Nurpur headwork and Shahdara headwork have design capacities of 4 MGD, 0.8 MGD, 0.7 MGD and 1.6 MGD, respectively. Currently, all waterworks are working on their maximum design capacity except for Korang water work (2.0 MGD), as per the CDA official record.

## Water demand and supply in twin cities

**Water demand and supply in Islamabad.** The Water Supply Directorate (CDA) is responsible for water supply services (production, treatment, conduction and distribution) in

Islamabad city. The CDA provides water and sewerage facilities to approximately 0.68 million people through 140,000 connections (of which 93% are residential and 7% are commercial). Currently, water supply coverage is above 95%, as almost all housing units are connected with a piped water supply per CDA records. The water supply is intermittent and available for 2–3 hours per day on average. At the domestic level, the occupants have constructed water tanks (underground and rooftop) to address water shortages to ensure water accessibility around the clock (24/7). Moreover, the bulk water supply from surface reservoirs is met (50–60%). However, it is worth mentioning that water metering is not imitated at the moment of the domestic water supply. Consequently, inhabitants are not sensitized or encouraged toward conserving valuable freshwater supplies. Additionally, water charges/tariffs are very nominal in the federal capital city (US$ 2.25–5.5/household/month). However, the water billing and collection efficiencies are 100% and 99%, respectively. As water tariffs are highly subsidized by the government in metropolitan cities, almost all city dwellers can afford water bills [39].

According to the findings of the Pakistan Social and Living Standards Measurement Survey (S3 Table ), in Islamabad, motor pumps (42%) constitute the main source of drinking water, followed by tap water (26%), filtration plants (17%), tankers (3%) and hand pumps (0%) (PSLM, 2019–20). According to data provided by the CDA, the per capita water demand is 60 GPCD (the maximum water consumption is 108 GPCD in the summer season), resulting in a cumulative domestic water demand of 72–84 MGD in Islamabad city. Additionally, the nondomestic water demand is recorded to be 70 MGD for commercial, other buildings, health and educational institutes. Therefore, the total water demand of Islamabad city is estimated to be 142 MGD, excluding private housing societies, slums and villages. The total water demand of ICT is registered as 175 MGD with the addition of rural areas, housing societies and slums. To meet this demand, the total water supply from surface water, groundwater and all other sources accounts for 65 MGD (S3 and S4 Tables). The details of the present production/consumption data are presented in Table 3. In addition, the CDA also operates a tanker service comprising 36 tankers to bridge the water demand and supply gap in Islamabad city at nominal charges. In addition, 37 water filtration plants based on tube wells have been installed in different sectors to cater to portable water needs in Islamabad city [40]. The findings reveal a shortfall of 77 MGDs in Islamabad, which are ostensibly met through unregistered private and domestic bore wells. Water loss of approximately 35% was reported by CDA due to faulty pipelines, unauthorized connections, old infrastructure and theft.

**Water supply in Rawalpindi city through WASA.** Similarly, water supply and sanitation services in Rawalpindi city are administered by the WASA, excluding areas under cantonment, Bahria Town, the DHA and private housing societies, covering an area of approximately 252 km². Currently, the WASA provides water facilities to approximately 1.9 million people with approximately 0.920 million water connections. Currently, the water supply coverage is above 85%. The civic agency has 127,019 consumers, of which 111,502 are domestic consumers and the remaining 13,506 are commercial consumers, as per the WASA record (Tables 3 and 4). In Rawalpindi city, a piped water supply is available for 8 hours per day, which is comparable to that of other metropolitan cities in the country, such as Karachi (4 hrs), Lahore (17 hrs), Faisalabad (8 hrs), Multan (8 hrs), and Peshawar (9 hrs) [41].

Currently, the per capita water demand in Rawalpindi city is 46.40 GPCD (the maximum daily water consumption is 83.52 GPCD on peak days in the summer season), which is much lower than the daily human requirements. The cumulative water demand in the WASA-served area is 64 MGD. The total water supply obtained from all three sources (Khanpur Dam, Rawal Dam and tube wells) is 51 MGD. These statistics reflect the demand and supply shortfalls of 13 MGDs, which are very challenging for civic agencies and are usually met by house boreholes and private water tankers. The water tanker service is composed of 3 tractor tankers, 21

**Table 4. Existing water supply scheme in the twin cities of Islamabad and Rawalpindi.**

| Description | Water and Sanitation Agency-Rawalpindi (R-WASA) | Rawalpindi Cantonment Broad (RCB) | Capital Development Authority (CDA) |
|---|---|---|---|
| Population (WASA Area) | 1.9 Million | 0.355 Million | 1.0Million |
| % Coverage | 85 | 80% | 95% |
| Total Consumer/water connections | 127,019 | 51,898 | 37898 |
| Total No. of Domestic Consumer | 111,502 | 51,351 | 34243 |
| Total No. of Commercial Consumer | 13,506 | 547 | 3650 |
| Planned Consumption | 40 GPCD | 30 GPCD | 60 GPCD |
| Rawal Dam | 10 MGD | ---- | 4* MGD |
| Khanpur Dam | 06 MGD | ---- | 51* MGD |
| Simly Dam | ---- | __ | 42* MGD |
| Tube wells | 40 MGD | --- | 47* MGD |
| Total Production | 51 MGD | 3.074 MGD | 60-64.54 (62) MGD |
| Average demand (2020-2021) | 64 MGD | 8.520 MGD | 176 MGD |
| Total peak production | 56 MGD | ---- | 84 MGD |
| No. of Tube wells | 400 | 58 | 206 |
| No. of Functional Tube wells | 264 | 58 | 206 |
| Water tanker/Bowzers | 26 | 09 | 36 |
| Sewerage Network | 35% | ---- | >90% |
| Sewerage treatment | 35% | ---- | 20% |
| Mini Filtration Plants | 150 | 48 | 37 |

Source: WASA Rawalpindi, 2021 http://wasarwp.punjab.gov.pk/AboutUs; Rawalpindi Cantonment Broad, (RCB), 2021 (rcb.gov.pk/en/water); CDA, 2021 (water supply directorate & Bulk water supply directorate) https://www.cda.gov.pk/projects/waterSupply.asp

*(installed capacity)

large and 2 mini water bouzers at nominal cost (domestic charges: Rs. 375; commercial Rs. 625 and tanker filling charges from WASA filling points: Rs. 250 per tanker). Additionally, nearly 150 filtration plants have been operated and maintained by WASA in Rawalpindi city to provide potable water to consumers. Water loss of approximately 40% was reported by WASA because of faulty pipelines, unauthorized connections, old infrastructure and theft.

**Water supply in Rawalpindi city through RCB.** The RCB is responsible for supplying water to the Rawalpindi Cantt area. Currently, RCB receives water through both surface and groundwater sources. The surface water supply is obtained through Khanpur Dam, whereas the groundwater supply comes from 58 deep tube wells installed in the cantonment area by the RCB. According to the records of the RCB, approximately 0.355 million people are currently served through all the sources. The total coverage is nearly 80% through 51,898 water connections, of which 51,351 are residential and 547 are commercial. The current water demand in the RCB jurisdiction area is 30 GPCD, with 54 GPCD maximum daily water demands on a peak day in the summer season. RCB is currently supplying only 3.074 MGD to meet domestic water needs against the total daily demand of 8.520 MGD, leaving behind the shortfall of 5.446 MGD. To overcome the issue of water shortages, the population is served through water tanker services (nine water bouzers) at nominal cost. In addition, RCB has installed and maintained nearly 48 filtration plants in the study area to provide safe and clean water to its consumers. According to the statistics published in the report of the state of Pakistani cities, in Rawalpindi city, piped water (46.6%) is the main source of drinking water, followed by motorized pumping (32.3%), filtration plants (13.8%), others (7.4%), tankers (0%) and hand pumps (0%). The existing water demand and supply in twin cities are summarized in Table 5.

As reported by concerned officials, another factor contributing to water demand-supply deficits is outdated water supply networks in twin cities. Leakages from the main water supply lines are common because thousand gallons of precious water are being lost on a daily basis. Consequently, 30–40% water losses were recorded in the study area. Moreover, water theft and unauthorized connections were also the main concerns.

The results of the present study clearly revealed that a large gap exists between water supply and demand in the Islamabad-Rawalpindi metropolitan area. Water demand (domestic, industrial and agricultural) is expected to increase further in the coming years as a result of rapid population growth, lifestyle changes, increases in the number of housing units, economic activities, the establishment of industries and urbanization. The findings highlighted that water demand has increased substantially in twin cities since the 1990s, which is consistent with rapid population and urban growth. The water demand was 102 MGD, 105 MGD and 119 MGD in 2001, 2003 and 2010, respectively, which reached 176 MGD in 2021 in Islamabad. Similarly, the urban water demand estimated for Rawalpindi city was 50 MGD and 56 MGD in 2001 and 2003, respectively, which reached 72.5 MGD in 2021. Consequently, twin cities have gradually transformed into water-scarce cities.

## Future water demand and supply projection from 2025–2050 in twin cities

The future water demand in twin cities was calculated on the basis of three future scenarios. The domestic water supply forecast results developed for the twin cities under the three scenarios are presented in supplementary material (S5–S7 Tables) and are presented in Figs 4 and 5 respectively. Under Scenario A in Islamabad city, urban domestic water demand will increase drastically from 90.99 MGD in 2025 to 193.27 MGD in 2040 and further rise to 319.34 MGD in 2050 if current demographic trends and distribution losses (35%) continue in the years to come. Similarly, the gap between demand and supply will also increase further from 48.74 MGD in 2025 to 151.02 MGD in 2040 and further increase to 277.09 MGD in 2050 in the capital city. Under Scenario B, increasing water losses from 30% to 10% from 2025–2050 can reduce the water deficit to 45.49 MGD in 2025 and 260.84 MGD in 2050 from the baseline. Under Scenario C, the water availability to the end user will improve further without water loss, and the water deficit can also be reduced considerably.

In Rawalpindi city, domestic water demand will increase drastically from 86.86 MGD in 2025 to 118.98 MGD in 2040 and further increase to 146.75 MGD in 2050 if current demographic trends and distribution losses continue in the future (Scenario A). Similarly, the gap between demand and supply will also increase further from 51.72 MGD in 2025 to 83.84 MGD in 2040 and further increase to 111.61 MGD during 2050 in the Rawalpindi metropolitan area. Under Scenario B, increasing water loss from 30% to 10% can reduce the water deficit to 49.01 MGD in 2025 and 98.09 MGD in 2050 from the baseline. Under Scenario C, the water availability to the end user will improve further without water loss, and the water deficit will also decrease significantly in 2025 (32.79 MGD) and 2050 (−92.68

**Table 5. Water demand and supply scenarios of twin cities in 2021.**

| Sr. No | Study Area | Per capita water demand (GPCD) | Total water Demand (MGD) | Total water Supply (MGD) | Deficit (MGD) |
|---|---|---|---|---|---|
| 1 | Islamabad City | 60 | 142 | 65 | 77 |
| 2 | Rawalpindi RWASA | 46.4 | 64 | 51 | 13 |
| 3 | RCB | 30 | 8.520 | 3.074 | 5.446 |

Source: CDA:2021; WASA:2021; RCB: 2021

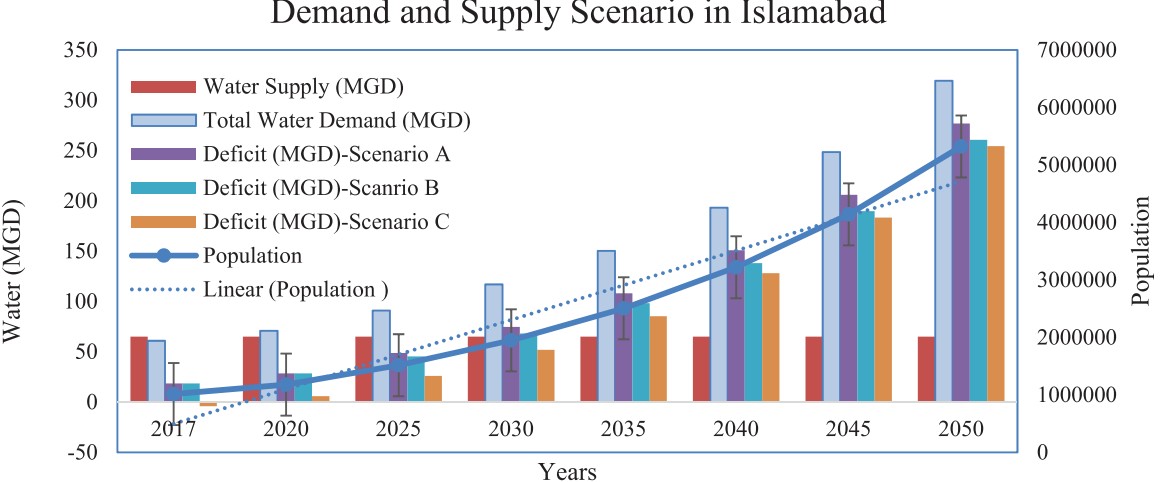

**Fig 4. Water demand and supply forecasting scenarios in Islamabad city from 2017–2050.**

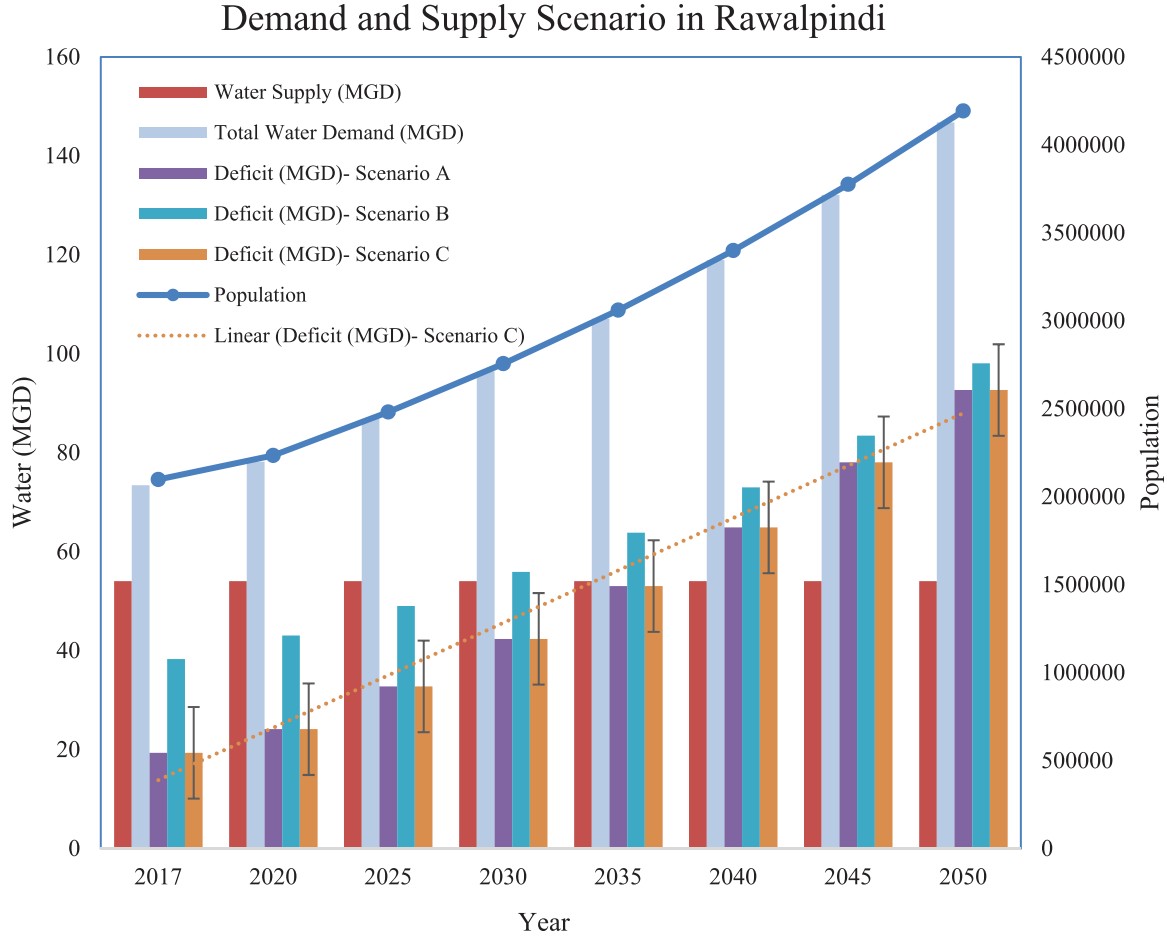

**Fig 5. Water demand and supply forecasting scenarios in Rawalpindi city from 2017–2050.**

MGD) compared with Scenarios A and B. In Scenario B, the deficit between demand and supply can be lessened if distributional losses are improved in twin cities by adopting different measures, such as water metering, improvement in infrastructure, reduction in water thefts and unauthorized connections. The findings of the present study are in good agreement with those of other water demand forecasting studies conducted in rapidly urbanizing regions of the world, such as North Carolina and South Carolina, USA, by Sanchez et al., 2020; Madurai city, India, by Vani et al., 2017; and Qatar, by Baalousha and Ouda, 2017 [8,11,12].

## Groundwater level decline

The groundwater table in twin cities is decreasing at an alarming rate because of unsustainable utilization of water, increased groundwater abstraction, and diminished water infiltration rates due to urban expansion, industrial and economic growth and an increasing population. In 1986, the water table in Islamabad was approximately 12 m (39.37 ft) beneath the ground surface, which further decreased to 30.50 m (100.06 ft) in the year 2003. The groundwater level further decreased to 35.70 meters (117.12 ft) during the next 12 monitoring years, i.e., 2003–2015. These trends revealed that during both monitoring years, i.e., 1986–2003 and 2003–2019, the water table declined at average rates of 1.09 m and 1.40 m per year, respectively. On average, groundwater is depleted at a rate of 1.7 m per year in the federal capital. The water table is predicted to drop to more than 46 m (151 feet) by the year 2025.

Similarly, in Rawalpindi city, the average water table depth is approximately 280 feet on the basis of the depth of the water table data obtained from the tube wells. As per the record of the WASA, the groundwater level decreased remarkably from 12 m (39.37 ft) to 45 m (148 ft) during the period of 1982–2001. At the current pumping rates, the water table decreases by an average of 1.83 m (6 ft) per year in Rawalpindi city. The condition is particularly alarming in the 46 old Union Councils (UCs), as reported by WASA. A maximum 20 meter (65.62 ft) drop in groundwater level was recorded in the Gulshanabad area. Additionally, according to data provided by CDA and WASA, the water table further decreased to 56 meters (183.72 ft) and 59 meters (193.57 ft) in Islamabad and Rawalpindi, respectively, from 2019–2020. The current trends clearly indicate that abstraction rates surpass recharge rates, undermining the safe yield of urban aquifers in twin cities. This decrease in groundwater level correlates well with the density of the distribution of pore wells in the study area [42]. The results pertaining to the lowering of table water and associated drying of aquifers corroborate well with the findings of previous studies conducted in twin cities and other parts of the country facing similar environmental risks (10,13–15)].

The forecasted results highlighted that increasing domestic water demand will pose a considerable challenge for increasing and maintaining water supply sources and infrastructure within twin cities. Moreover, as urban water demand increases in twin cities in response to numerous demographic, urban and economic factors, it is imperative to understand the elements that can reduce future water demand or increase the water supply to achieve urban water security. Adequate policy formulation for regulating/abstracting groundwater, technological improvements for reducing distributional losses, water theft, unauthorized connections, training staff, educational campaigns for mass awareness and revising pricing structure or eliminating water subsidies could have contributed significantly to the more efficient use of fresh water resources in the rapidly growing metropolitan area of Islamabad-Rawalpindi. The present study further highlights the need for integrated water management and integrated land use planning/modeling that provide evidence-based solutions for concerned policy makers and urban town planners.

## Key drivers for increasing water demand

**Urban population growth.** Cities are considered the engine of economic development. Therefore, the phenomenon of urbanization is regarded as both an opportunity and a challenge for town planners and decision makers [43]. Although the proportion of the urban population is increasing rapidly worldwide, the rate is much faster in developing countries. Therefore, estimation of past, present and future demographic trends is a key driving factor for providing safe, accessible and affordable water to communities[10]. Hence, demographic features are crucial for the sustainable management of water resources in any region, as they have direct micro- and macro-environmental implications for resource availability. To forecast present and future water demand, population growth trends for twin cities were plotted from 1951–2017 (Fig 6).

The findings revealed that Islamabad and Rawalpindi conurbations experienced dramatic increases in annual population growth rates of 5.15% and 6.50%, respectively, which were attributed to natural and migration factors. Twin cities are categorized among the top ten cities with populations above five million and are therefore considered "urban heavy weights" owing to their population and size. The census data revealed that the population increased from 95,940 to 2,006,572 in Islamabad and 2,37,219 to 2,098,231 in Rawalpindi from 1951–2017. According to the 1998 census, the populations of Islamabad and Rawalpindi were 805,235 and 1,409,768, respectively [44]. Furthermore, the intercensal record (1998–2017) highlighted that the population of twin cities has increased twofold-fold during the last two decades. The collected data revealed that the increase in population is not steady. The fastest growth (5.19%) was observed during the period of 1981–1998 due to the influx of Afghan refugees during the USSR-Afghan War. In addition, internal migration toward twin cities contributed by terrorism in the tribal areas of FATA, KPK and Balochistan can also be correlated with higher migration rates in these areas. Finally, natural disaster-induced migrations (such as the Great Kashmir Earthquake of 2005 and the devastating flood of 2010) also directed a significant number of people in twin cities, particularly Islamabad [10]. The migration trends found in the study area are presented in Table 6. Migration-induced population growth is trickier for planners, as instead of steady growth, it results in a sudden influx of people in an area. It is anticipated that an increase in the frequency of catastrophic events in other areas

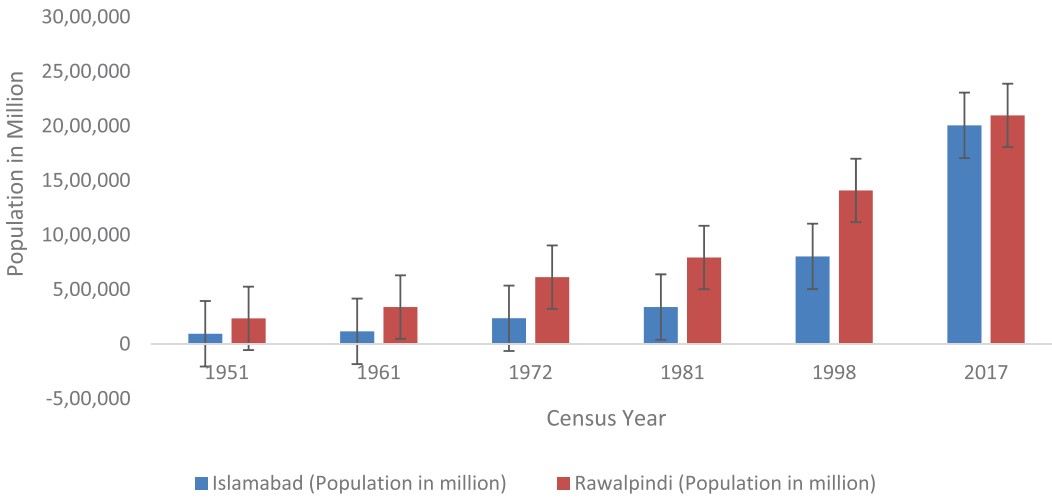

**Fig 6. Population growth trends in the twin cities of Islamabad & Rawalpindi.**

can result in an unexpected increase in the population of twin cities, burdening the already dwindling water resources of this region.

The metropolitan areas of Islamabad and Rawalpindi experienced high rates of urbanization. According to records of census reports, the proportion of the urban population increased significantly from 32.9% in 1972 to 65.6% in 1998 in ICT. However, in ICT, a considerable decline has been recorded in the proportion of the urban population (with 50.60% urban dwellers in 2017) compared with the 1981–1998 censuses. This declining trend in the urban population clearly demonstrated that major growth occurred in the rural counterpart of the capital city (Fig 7). Most of the population shifted and settled in numerous housing societies established along the Islamabad expressway and on the rural–urban fringe, resulting in urban sprawl. According to records of CDA, the population of Islamabad city is burgeoning at a rate of 4.91% per year and is predicted to reach the figure of 4 million by the year 2030 [40,45]. Similarly, the proportion of the urban population slightly decreased from 80.90% in 1972 to 77.30% in 1998, which further decreased to 48.89% in Rawalpindi city.

Islamabad and Rawalpindi conurbations are placed among the most heavily populated areas of the country, with a combined population density of 3000 $km^2$, as per the 2017 census record. Previous census records revealed that the population density increased notably from 106 $km^2$ to 2215 $km^2$ in Islamabad and from 915.90 $km^2$ to 8,100 $km^2$ in the Rawalpindi metropolitan area from 1951–2017 [46] (Fig 8.). A plausible explanation links higher migration rates to economic and developmental activities in urban areas, particularly in developing countries such as Pakistan [10]. Therefore, internal migration plays a crucial role in reshaping the demographic landscape and socioeconomic, cultural, political and environmental facets of both native and host regions [44]. Islamabad-Rawalpindi metropolitan conurbations received a massive influx of approximately 1,063,576 migrants from rural areas, with an urban growth rate of 3.46% per the 1998 census record (S2 Table ).

**Growth in housing units.**  Housing is among most elementary human needs; however, at the same time, its construction causes land use/land cover changes, i.e., the conversion of land cover to non-vegetative, impermeable surfaces. The summarized data pertaining to housing units in twin cities are presented in Fig 9. There are approximately 336,182 and 470,588 housing units registered in Islamabad and Rawalpindi, respectively, according to the National Housing and Population Census, 2017. Furthermore, according to the 2017 census, the urban populations of Islamabad and Rawalpindi are distributed in 170,936 and 470,588 housing units, respectively, whereas 86,575 and 212,429 housing units, respectively, are distributed according to the 1998 census. The findings revealed that housing units increased remarkably during the intercensal period 1998–2017. However, Rawalpindi city has experienced a noteworthy increase in housing units (258,159) compared with Islamabad city (84,361) due to the development of mega-housing projects such as Bahria Town (Phase I-VIII), Green Valley, DHA, Bahria Enclave, Lake View city, and the Wallaiyat complex in the aforementioned city. There has also been a growth of unplanned settlements in the city [47]. In 1998, the share of urban households was 69.2%, which has since declined because of the high cost of living in regulated urban settlements. The average household size displayed a declining trend from

**Table 6.  Migration trends in the metropolitan conurbation of Rawalpindi-Islamabad (% age).**

| Study Area | Native | Migrated | Intra Migration | Inter Migration |
|---|---|---|---|---|
| Islamabad | 64 | 36 | 17 | 19 |
| Rawalpindi | 85 | 15 | 7 | 8 |

Source: Pakistan Social and Living Standard Measurement Survey 2019-20

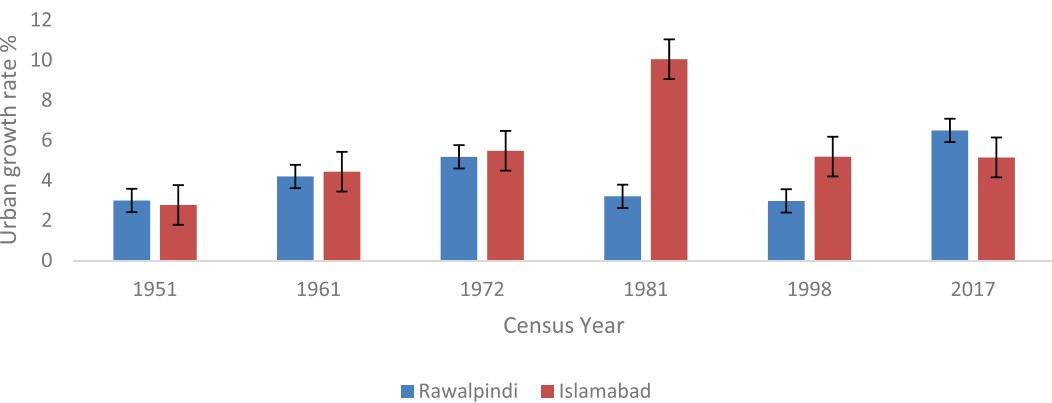

**Fig 7. Urban growth in the Twin cities of Islamabad & Rawalpindi.**

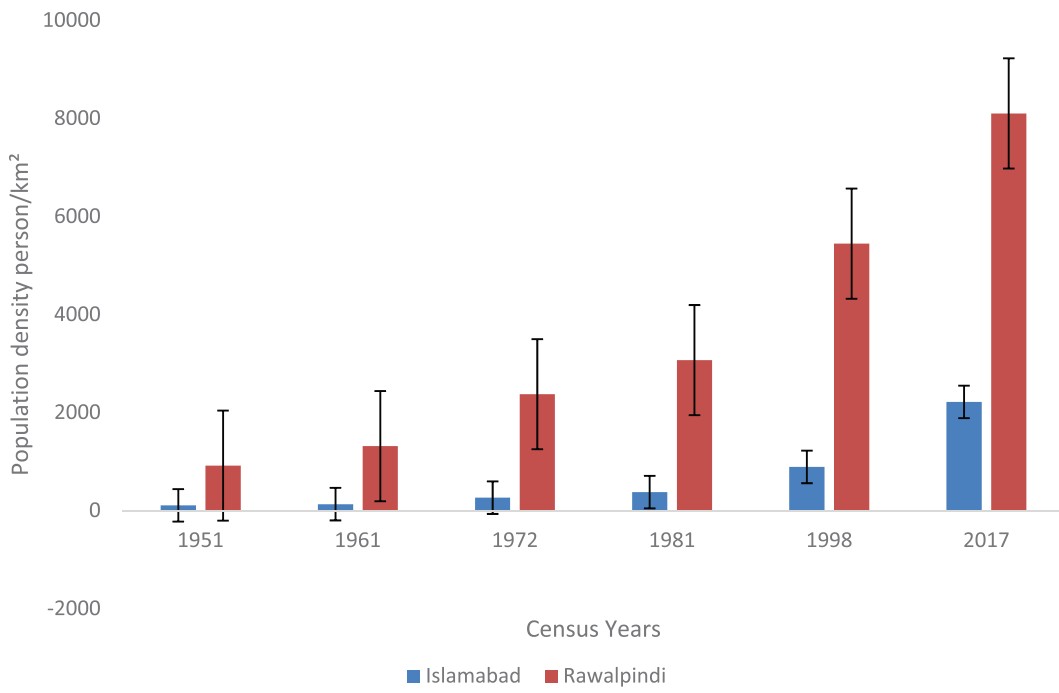

**Fig 8. Population density trends in Islamabad and Rawalpindi city from 1951–2017.**

6.2–6.0 persons per household in Islamabad and from 6.64–5.8 persons per household in Rawalpindi city from 1998–2017 [48]. The increasing growth rate and declining household size clearly demonstrate that housing demand will increase further with time. It is anticipated that this trend will require horizontal and vertical expansion of urban areas, as more space is needed to accommodate rapidly growing populations by overwhelming rural areas and rural-urban fringes [49]. These interventions transform open spaces into built-up areas and land, putting urban ecosystem services at risk. All these modifications can cause changes in recharge rates, reduce infiltration and increase runoff, consequently contributing to the drying of surface and subsurface water resources in urban areas. The proliferation of manmade infrastructure coupled with exponential population growth in urban agglomerations poses

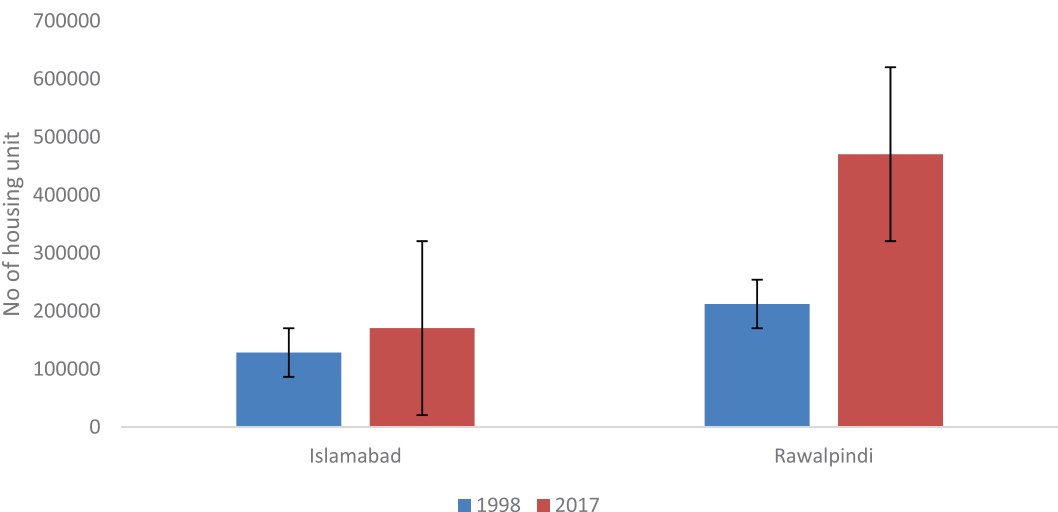

**Fig 9. Number of housing units in twin cities of Islamabad & Rawalpindi from 1998–2017.**

several environmental challenges, including water demand-supply deficits [50]. The various types of housing units present in twin cities are given in S2 Table.

## Land use/land cover change (LULCC) and its impact on the ground water table

Current research is explicated by relating the land use/land cover change (LULCC) of twin cities with declining ground water levels. Land use refers to the purpose for which people utilize landscape resources, while land cover is the natural and artificial component of the landscape. LULCC refers to the natural and anthropogenic modifications that humans carry out to cater to their needs [10,49].

Industrial expansion, economic growth and advancements in lifestyles are the key drivers of LULC changes around the globe. The most commonly observed pattern of LULC change was a reduction in vegetation cover and an increase in built-up areas as urban and economic progress in different parts of the world. Cities are dynamic, and the changes that take place are inevitable. The modification of urban areas can be attributed to numerous factors, but "population growth" is considered the prime factor behind urbanization and urban sprawl. Indeed, rapid population growth places a high demand on natural resources and urban ecosystem services, which ultimately leads to the degradation of the urban environment [10,27,50,51].

The 20-year LULC data from 1999–2019 from Rawalpindi Islamabad were analyzed, and the results are illustrated in Figs 10 and 11 respectively. The analysis of LULC changes (Table 7) clearly revealed that twin cities experienced rapid urbanization and urban sprawl during the study period, i.e., 1999–2019. LULC is rapidly changing in the urban centers of Rawalpindi and Islamabad in response to socioeconomic infrastructure development. Beautiful environmental settings; better access to education, health, and transportation facilities; employment opportunities; and good salary packages attract many migrants to twin cities. Consequently, urban and peri-urban expansions have taken place at unprecedented rates.

The findings of the present study clearly revealed that the LULC types changed significantly from 1999–2019 in twin cities. The results demonstrated that the maximum change

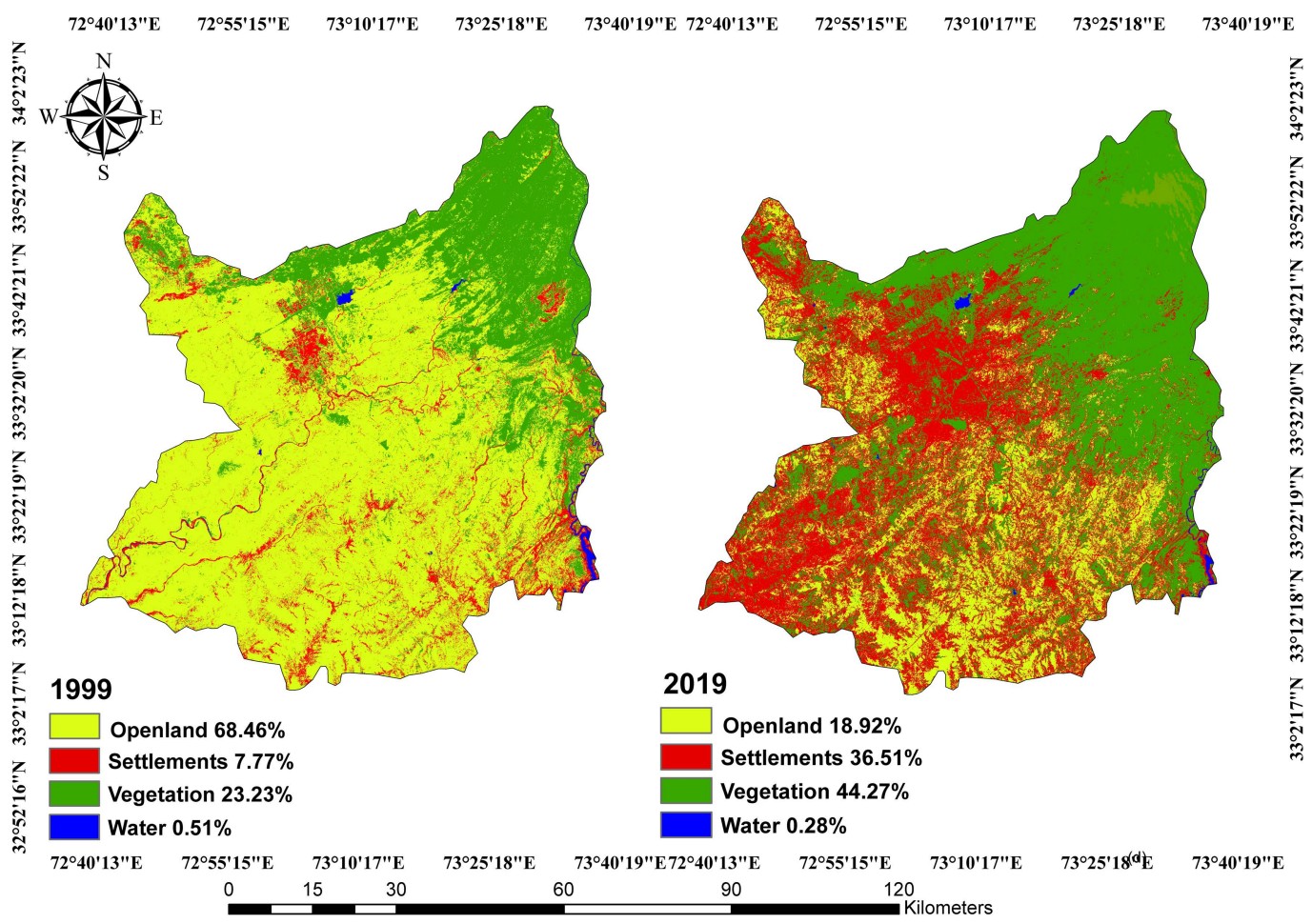

**Fig 10. Land use/land cover maps of Islamabad-Rawalpindi Metropolitan (1999) and (2019).**

was rapid decline in open land, which has been replaced by settlements. Specifically, the area of open land declined by 49.54%, whereas the rate of increase in settlements during the study period more than doubled. In the previous 20 years, the total vegetation increased from 23.23% to 44.27%, showing a net increase of 21.04%, whereas water resources experienced a slight reduction of 0.231%. From 2001–2018, the LULC change in Dehradun city, India, demonstrated a 73.53% increase in the built-up area and a 49.67% decrease in agricultural land over 17–18 years [52]. Similarly, the LULC changes in Bauchi, a city in Nigeria, indicated 138% increases and 64.1% decreases in the built-up area and farmland, respectively, over a 10-year time span [53]. These land use/land cover changes have highly impacted the aquifer system of twin cities, which is quantitatively in line with over-extraction by the expanding population. The groundwater level decreased by approximately 1.5 m annually from 1998–2007 due to uncontrolled abstraction. However, from 2007–2009, the reduction in the groundwater level further spikes to 3 m in annual decline and 4.3 m in annual decline from 2009–12, which is alarming. If the current trend of decline continues in the coming years, the groundwater aquifer will shrink to a frightening extent. For this reason, groundwater has become an unreliable source of urban water for twin cities. City dwellers are at severe risk of water deficiency in the near future. Some of the posh areas of Rawalpindi and Islamabad have recorded the highest ecological/water footprint.

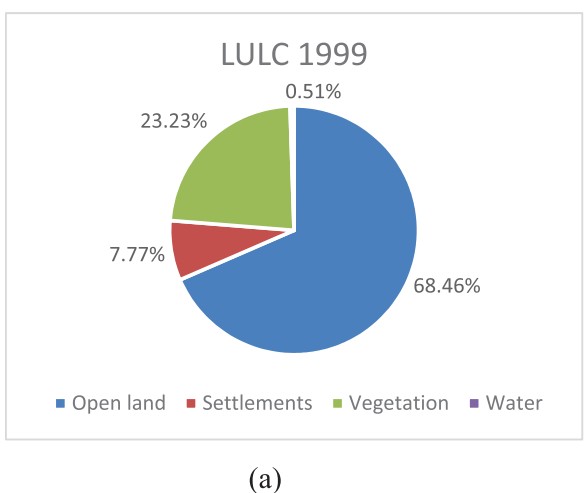

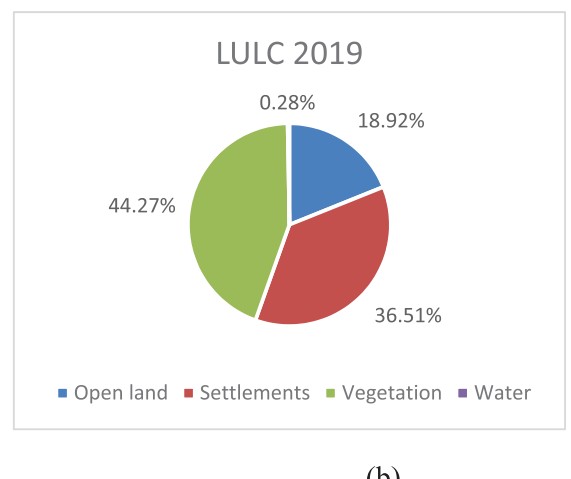

(a)

(b)

**Fig 11. LULC of study area for the year 1999 (a) and 2019 (b).**

Table 7. Change in landuse land cover (LULC) Classes in Islamabad-Rawalpindi City (1999–2019).

| Sr. No | Parameters/Land cover classes | 1999 % | 2019 % | Change % |
|---|---|---|---|---|
| 1 | Open land | 68.46 | 18.92 | 49.54 ↓ |
| 2 | Settlements | 7.77 | 36.51 | 28.74 ↑ |
| 3 | Vegetation | 23.23 | 44.27 | 21.04 ↑ |
| 4 | Water | 0.51 | 0.279 | 0.231 ↓ |

Quantitative analyses of the impacts of LULC change on groundwater have been performed. The results revealed that the large influx of the population toward cities has a significant impact on the city's ecological resources and has altered its land use patterns, such as topography and water usage. Large-scale permeable land is replaced by impermeable land because of the maximum utilization of concrete, asphalt and metal-like materials. This conversion of open space into a built environment largely influences surface and groundwater resources both quantitatively and qualitatively, such as the rates of water recharge, topography, land cover, rainfall patterns and pollution. Additionally, reductions in the groundwater recharge rate and increases in urban flooding are strongly connected with LULC patterns. LULC changes modify the groundwater recharge mechanism through altering its hydrologic characteristics, such as infiltration, evapotranspiration, water storage and retention [30,54,55].

## Limitations of the study

An unfortunate coronavirus outbreak occurred during the research duration. Owing to lockdown, we face severe difficulties in conducting field surveys, household community surveys and interviews with key stakeholders. Therefore, to address this situation, an online Google questionnaire was developed that can be reached at the link (https://docs.google.com/forms/d/e/1FAIpQLSeWqoiMzOi9QN8WoEJajSQikL_ooMktda5xDbrum6hzWxwsw/viewform?usp=pp_url). Unfortunately, the responses on the online Google form were unsatisfactory; thus, unwantedly, we skipped the community household survey component due to lockdown. As soon as the situation improved, we completed our questionnaire survey from concerned civic agencies, and 20 in-depth interviews were conducted. Moreover, the current research does not focus on the analysis of the impact of climate change on the water resources

of the Islamabad-Rawalpindi metropolitan area. The present study focused primarily on the estimation of domestic water demand in the study area, and nondomestic water demand was not addressed in this study. The present study focuses on the analysis of existing water resource use and supply, water demand–supply imbalances, key driving factors and LULC changes via simple equations for population projections and water demand–supply forecast scenarios.

### Future work

The present research study suggests the estimation of nondomestic water demand (such as industrial demand, agricultural demand, ecological demand and rural demand) in the Islamabad-Rawalpindi urban agglomeration in the future.

Moreover, the current research also proposed a study related to the impact of climate change on the water resources of the Islamabad-Rawalpindi metropolitan area.

For future work, the present study suggests the prediction of water conditions via a statistical or economic model to validate the results of the present investigation.

## Conclusions and recommendations

The water demand-supply imbalances associated with LULC changes, urbanization and demographic factors have been a well-known phenomenon across the globe particularly in the rapidly growing developing countries. The increasing population in twin cities makes them very diverse and urbanized, which increases the total water demand for the domestic, agricultural and industrial sectors. Therefore, present study was undertaken to examine the status of water resources, existing and projected domestic water demand and supply (in the context of business as usual, gradual improvement in water losses and without water losses) scenarios up to 2050, associated LULC and the key factors influencing water demand in the Islamabad-Rawalpindi metropolitan area. Digital image processing tool was employed to delineate LULC classification of the twin cities. Secondary data was collected from concerned departments to study water demand-supply, demographic trends and water-table depth. The LULC analysis of the study area depicted that the open land reduced significantly from 68.46% to 18.92% due to its replacement by human settlements during 1999-2019. Meanwhile the area under built environment also increased significantly from 7.77% to 36.51%. However, increase in the forest cover and water was found to be insignificant during last two decades. The conversion of permeable land into impermeable land implies a massive impact on groundwater recharge via the rapid flow of rainwater, which causes urban flooding during heavy rains and slows the process of infiltration. Consequently, groundwater level has declined from 22.8 to 56 meters in Islamabad and from 37.8 to 59 meters in Rawalpindi city during 1999-2019. The lowering of water table is occurring at rates of 1.7 m and 1.83 m per year in Islamabad and Rawalpindi respectively causing the deepening and drying of wells in different parts of twin cities. This declining trend is strongly associated with the density of tubewells and bores installed, demographic trends and land transformations. The condition is particularly alarming in the old Union Councils of Rawalpindi city compared with Islamabad, where the maximum dropdown was observed in Gulshanabad. The findings of present study indicated that domestic water demand also increased significantly in response to escalating population, urban growth, economic and industrial progress in twin cities during 1999-2017 and projected to grow further by 2050. Currently, water demand and supply deficit of 52 MGD and 18.50 MGD has been recorded for Islamabad and Rawalpindi respectively which is expected to grow further in the coming years and hence quite alarming for civic agencies. Therefore, the concerned civic agencies face challenges in meeting consumer

water demand because of the aging of infrastructure, water losses and theft, unauthorized connections, lack of institutional capacity, lack of finance, growing population, urbanization and associated LULC. The existing water supply systems of twin cities are largely based on groundwater, as supplies from surface water resources (Rawal, Simply and Khanpur Dams) are intermittent and inconsistent due to variations in rainfall, silt deposition and evaporation losses. Accordingly, water shortfalls have remained significantly greater, although the water supply capacity has also increased noticeably. The findings clearly indicated that expanding urban infrastructure is directly involved in the deterioration of the groundwater budget. The present study concluded that with the current rate of groundwater abstraction and LULCC, aquifers in twin cities will become nonproductive, making the study area highly water deficient.

The scenario results revealed that the Islamabad-Rawalpindi metropolitan area needs more water for growing populace in all three scenarios. However, in Scenarios B and C, the deficit between demand and supply can decrease if distributional losses are improved in twin cities by up-gradation of the rusted transmission lines and controlling theft. Furthermore, regularizing and controlling groundwater abstraction, rainwater harvesting, groundwater recharge, water consumption-based tariff systems, public awareness campaigns and minimizing water theft are the key steps to be implemented on an immediate basis. In this context, these water demand–supply data (present and forecast) provide a baseline to assist decision makers and urban planners in sustainably managing water resources in twin cities and serve as a reference for other metropolitan areas.

The forecast scenario presented in this study reflects the limitations of climate change and seasonal variations. Therefore, future studies should focus on estimating seasonal variations in water demand in connection with climate change. In addition, there is a dire need to develop new water sources in twin cities, most preferably those based on surface water or the combined use of both groundwater and surface water, to meet the growing present and future water demands for sustainable urban growth. Integrated water resource management may be adopted by twin city decision makers, policy makers and water resource managers to resolve the issue of water shortages. The adoption of different water-saving measures to guide the sound growth of demand and increase the relative carrying capacity of water resources is important in the water supply chain to effectively mitigate the shortage of clean freshwater. It is hoped that current study will provide valuable information to concerned stakeholders to take robust measures to address repercussions of unplanned urbanization and LULC to ensure sustainable water supply to the urban populace.

## Ways toward securing the future of water

To improve the water demand and supply scenario in twin cities, there is a dire need to develop and implement comprehensive water resource management plans for twin cities that should cover the following key measures:

i.    Unplanned urbanization needs to be controlled, and sustainable urban planning/designs need to be implemented in new urban projects.

ii.   Water loss (currently 30–35%) can be reduced by replacing rusty and leaky pipes, monitoring and preventing leakage, and controlling water theft. A reduction in water loss should be promoted for water-efficient technologies and appliances such as leakage recognition kits for bathrooms, sensor taps, and low-flow taps in homes.

iii.  Rainwater harvesting structures, including recharge wells, water reuse systems and tanks, should be made mandatory in new housing schemes/society by city managers and

policy makers. The strict enforcement parliamentary bill passed in April 2023 regarding mandatory rain water harvesting in new buildings.

iv.  There is an immediate need to control groundwater abstraction by formulating legislation/policies for the installation and operation of tube wells and borehole wells in the study area. Appropriate and strict actions must be taken by concerned municipalities to discourage the use of suction pumps on public water supply lines.

v.  Piezometers should be installed in at least one tube well located in each sector in Islamabad and at $^a$ distance of 2 km$^2$ in Rawalpindi city to monitor groundwater levels and dropdowns as well as to accurately discern groundwater flow pathways. In addition, appropriate tube well spacing is a major factor that can prevent the dropdown of the water table and keep it at a minimal level.

vi.  Water metering and billing systems should be made mandatory in all urban centers to discourage exploitation and waste.

vii.  Water conservation behavior should be promoted through awareness campaigns in schools, colleges and universities and through the use of print and electronic media.

viii.  To ensure a consistent urban water supply, the storage capacity of existing water reservoirs needs to be enhanced through integrated catchment area management. Small and mini dams such as the Chirah dam, Golra dam and community pond at F-9 Park need to be constructed without further delay by the government to meet the growing water demand in twin cities on a sustained basis.

ix.  Future studies should focus on estimating seasonal variations in water demand in connection with climate change.

## Supporting information

**S1 File. Questionnaire.**
(DOCX)

**Table S2. Demographic record of twin cities of Islamabad-Rawalpindi (1951-2017).**
(DOCX)

**Table S3. Various types of housing units in the study area (% age).**
(DOCX)

**Table S4. Major sources of drinking water in households (% age).**
(DOCX)

**Table S5. Water demand for a single house having 4-person accommodation in Islamabad City.**
(DOCX)

**Table S6. Water demand and supply forecast in Islamabad City from 2017–2050.**
(DOCX)

**Table S7. Water demand and supply forecast in Rawalpindi City from 2017–2050.**
(DOCX)

**Table S8. Key stakeholders.**
(DOCX)

## Acknowledgments

The first author would like to appreciate the assistance of Mr. Kamran Ahmed and Mr. Rehan Hameed Rana in data collection from concerned public departments.

## Author contributions

**Conceptualization:** Muhammad Ashraf, Naveed Iqbal Gondal.

**Data curation:** Sidra Aman Rana, Sadia Rahman.

**Formal analysis:** Sidra Aman Rana, Nadia Akhtar.

**Methodology:** Sidra Aman Rana, Syeda Maria Ali.

**Supervision:** Syeda Maria Ali.

**Writing – original draft:** Sidra Aman Rana.

**Writing – review & editing:** Syeda Maria Ali.

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
