## [Decision Letter · Decision Letter 0]

7 Jul 2023

PONE-D-23-14511Estimation of domestic water demand -Supply Scenario and its key driving factors in Islamabad-Rawalpindi Metropolitan Area, Pakistan.PLOS ONE

Dear Dr. Ali,

Thank you for submitting your manuscript to PLOS ONE. After careful consideration, we feel that it has merit but does not fully meet PLOS ONE’s publication criteria as it currently stands. Therefore, we invite you to submit a revised version of the manuscript that addresses the points raised during the review process.

We look forward to receiving your revised manuscript.

Kind regards,

Muhammad Tayyab Sohail

Academic Editor

PLOS ONE

Journal Requirements:

3. We note that Figure 1 in your submission contain map images which may be copyrighted. All PLOS content is published under the Creative Commons Attribution License (CC BY 4.0), which means that the manuscript, images, and Supporting Information files will be freely available online, and any third party is permitted to access, download, copy, distribute, and use these materials in any way, even commercially, with proper attribution. For these reasons, we cannot publish previously copyrighted maps or satellite images created using proprietary data, such as Google software (Google Maps, Street View, and Earth). For more information, see our copyright guidelines: http://journals.plos.org/plosone/s/licenses-and-copyright.

(1) You may seek permission from the original copyright holder of Figure 1 to publish the content specifically under the CC BY 4.0 license.  

**Additional Editor Comments:**

Your present paper is not well presented and you need to add more data and analysis. please read following articles. 

Reviewers' comments:

Reviewer's Responses to Questions

**Comments to the Author**

1. Is the manuscript technically sound, and do the data support the conclusions?

Reviewer #1: Partly

Reviewer #2: Partly

2. Has the statistical analysis been performed appropriately and rigorously? 

Reviewer #1: Yes

Reviewer #2: Yes

3. Have the authors made all data underlying the findings in their manuscript fully available?

Reviewer #1: Yes

Reviewer #2: No

4. Is the manuscript presented in an intelligible fashion and written in standard English?

Reviewer #1: Yes

Reviewer #2: Yes

5. Review Comments to the Author

Reviewer #1: The present paper titled “estimation of domestic water demand -Supply Scenario and its key driving factors in Islamabad-Rawalpindi Metropolitan Area, Pakistan” can be considered for publication in a prestigious journal like POLSONE with major revision. My specific comments for authors to address are given below. The authors need to carefully address all comments before the paper can be fully accepted for publication.

1. Many people around the world suffer from clean water scarcity. The present paper focus on demand and water supply in major connected cities of Pakistan. Though the work done is of comprehensive nature but still more work is required to justify the findings of the study. Urban expansion is compromising the water availability to justify this statement with change in Land use land cover in the area. Image acquisition is not hard when a freeware data set is available on LANDSAT. I would suggest acquiring time series data in order to see the trend and verify the results in terms of land use and land cover change in the area. At least a change over a period of 20 years interval would work and give better results when change in population has been reported.

2. What data sources were used to estimate domestic water demand in the study?

3. Are the data reliable and representative of the population in the Islamabad-Rawalpindi Metropolitan Area?

4. How was the water demand calculated?

5. Did the methodology consider factors such as population growth, household size, and more importantly seasonal variations in water usage?

6. Were any specific statistical techniques or models used to estimate water demand? If so, can you provide more details about their application and validation?

7. What are the identified key driving factors influencing domestic water demand in the Islamabad-Rawalpindi Metropolitan Area?

8. How were these factors determined or selected for analysis?

9. Were the driving factors analyzed individually or in combination to understand their relative contributions to water demand?

10. Can the authors provide insights into the magnitude of each factor's impact?

11. Were there any unexpected or significant findings regarding the driving factors?

12. Did the study reveal any previously unexplored factors that could influence water demand?

13. Did the study consider the existing water infrastructure, sources, and their capacity to meet the projected water demand?

14. Were there any limitations or challenges identified in the current water supply system or potential leakages in the institutional arrangements that could be improved?

15. Were potential solutions or strategies proposed to address any shortfalls or gaps?

16. How does the estimated water demand align with the existing or planned water supply infrastructure in the Twin Cities?

17. What are the implications of the study's findings for water resource management in the Islamabad-Rawalpindi Metropolitan Area? How can the results inform policy decisions or water conservation efforts?

18. Were any specific recommendations provided to policymakers, local authorities, or water management organizations based on the study's outcomes?

19. How do the findings of this study contribute to the existing literature on domestic water demand estimation and its driving factors, particularly in the context of urban areas in Pakistan?

20. What are the main limitations or potential sources of error in the study's methodology or data that might affect the accuracy or generalizability of the findings?

21. I suggest authors to clearly mention the potential areas for future studies related to domestic water demand estimation or management in urban areas?

22. The study did not consider climate change as a main driving factor in water scarcity, was there any particular reason for not considering it? If so, please mention it or I strongly suggest authors to add climate change, demographic shifts, seasonal variation in water utilization which could impact water demand in the Islamabad-Rawalpindi Metropolitan Area?

23. I strongly advise authors to read the current development made in the policy direction related to the Twin cities where the authorities have passed a parliamentary bill for rainwater utilization in all commercial and residential buildings. This might provide stronger arguments and insights when discussing research results.

Reviewer #2: Comments to authors

I had the opportunity to review your paper “Estimation of domestic water demand -Supply Scenario and its key driving factors in Islamabad-Rawalpindi Metropolitan Area, Pakistan.” The objectives of the study were: 1) to assess current and projected water demand and supply scenario in twin cities; 2) to relate demographic and urban trends as well as the density of tube wells with water table decline in twin cities.

It is very interesting work and has some merits. However, I have some comments and questions that the authors should address them clearly and scientifically.

1- The introduction

- The instruction is too broad. The authors should review and identify clearly (1) the revolution of domestic water demand supply and (2) the key driving factors of high demand and scarcity of domestic water.

- Also, the introduction would be more interesting if the authors addressed how water demand and scarcity negatively affected the socioeconomic development of a country, esp. Pakistan?

2- Methods and Materials

- The data collection procedure was not clearly described. For example, L152: The authors explained that the data was acquired from various organizations through extensive interviews. If so, the authors should indicate the number of respondents who were interviewed and who they were? What questions were used to interview? These are important!

3- Results

3.1. Demographic profile of the study area: What is the main point of this section that the authors want to provide to readers? I think this part should be moved to combine with 2.1.

- L233-234: …There are approximately 336,182 and 470,588 234 housing units registered in Islamabad and Rawalpindi, respectively… in what year? The authors should add the year to this growth.

- These two sections—3.3.1 and 3.3.2. are more like literature/a review of existing documents/reports

- Likely, 3.4 is more about reviewing documents/reports.

- I saw only two sentences in L365-366 and L389-390 that the authors likely reported some ideas/findings from the survey? However, it is inconsistent and unclear between what is mentioned in the methodology and these parts. In the methodology, the authors said that they conducted intensive interviews in the methodology, and here in the result, the authors said, “…reported by respondent during field survey.” So what exactly had the author done? Meanwhile, there was no other part of the result section explaining the results/findings from the interview or survey. I doubt it!

4- Conclusion

- L480-484: How did the authors come to that conclusion? There was no clue explaining this concern in the result section.

- Likely in 4.1. Ways toward secure water future. The authors gave many suggestions, but most are from the authors' ideas/perspectives rather than from this research findings. For example, “(i) …an immediate need for formulation of appropriate groundwater 519 legislation/policy….” There is not a sentence talking about the policy gap or need in the research; how did the authors suggest “an immediate need”? It seems out of what the research is doing. At least the authors discussed somewhere in the research about the policy and law before coming up with this suggestion. Specifically, I saw the terms “water theft, unauthorized connections…” which are things that the authors should suggest some actions be taken in order to improve the water supply or reduce the water deficit.

- More importantly, the authors should discuss and suggest what the authors proposed in Scenarios A, B, and C, which are in the scope of the study. If scenarios B and C are the best, how to implement them?

6. PLOS authors have the option to publish the peer review history of their article (what does this mean? ). If published, this will include your full peer review and any attached files.

**Do you want your identity to be public for this peer review?** For information about this choice, including consent withdrawal, please see our Privacy Policy .

Reviewer #1: **Yes: ** Wahid Ullah

Reviewer #2: No

---

## [Author Response · Author response to Decision Letter 1]

21 Aug 2023

Estimation of domestic water demand -Supply Scenario and its key driving factors in Islamabad-Rawalpindi Metropolitan Area, Pakistan.

Manuscript ID: PONE-D-23-14511

Reply – Editor Comments

Dear Editor,

We would like to thank you for your valuable comments about our manuscript titled “Estimation of domestic water demand-supply scenario and its key driving factors in Islamabad-Rawalpindi Metropolitan Area, Pakistan” (ID: PONE-D-23-14511). Those comments are all valuable and very helpful for revising and improving our paper, as well as the important guiding significance to our research. We have studied comments carefully, incorporated all necessary corrections which seemed technically sound, and on logical grounds. We hope it meets the requirements for your kind approval. All changes in the paper are marked in track change mode in the submitted version of the document. The following are responses to the comments.

Note: Prefix “C” means Comment, Prefix “R” means Response

C-1. Please ensure that your manuscript meets PLOS ONE's style requirements, including those for file naming.

R-1. Needful has been done.

C-2. We note that you have stated that you will provide repository information for your data at acceptance. Should your manuscript be accepted for publication, we will hold it until you provide the relevant accession numbers or DOIs necessary to access your data. If you wish to make changes to your Data Availability statement, please describe these changes in your cover letter and we will update your Data Availability statement to reflect the information you provide.

R-2. All the repository information is provided in supplementary tables and excel sheets.

C-3. We note that Figure 1 in your submission contain map images which may be copyrighted. We require you to either (1) present written permission from the copyright holder to publish these figures specifically under the CC BY 4.0 license, or (2) remove the figures from your submission.

R-3. Figure 1- Study Area Map was developed by the author. Background image layer has been acquired from USGS (freely available), therefore no copyright issue.

C-4. Please include a separate caption for each figure in your manuscript.

R-4. Authors already gave separate caption for each figure in the manuscript. Needful has been done.

C-5. Please include captions for your Supporting Information files at the end of your manuscript, and update any in-text citations to match accordingly.

R-5. Supporting information given is incorporated in in-text citations to match accordingly and also give information at the end of manuscript. Needful has been done.

Additional Editor Comments:

C-6. Your present paper is not well presented and you need to add more data and analysis. please read following articles.

• 10.5004/dwt.2020.25119

• 10.5004/dwt.2019.24156

R-6. Thank you so much for your suggestions to improve quality of manuscript. These articles are already reviewed and one is incorporated. More data analysis has been done to improve manuscript like LULU. However, we have another submitted paper on water quality assessment using drinking and irrigational water quality indexes. Therefore, this aspect cannot be included in this manuscript.

Estimation of domestic water demand -supply scenario and its key driving factors in Islamabad-Rawalpindi Metropolitan Area, Pakistan.

Manuscript ID: PONE-D-23-14511

Reply – Reviewer #1

Dear Reviewer,

We would like to thank you for your valuable comments about our manuscript titled “Estimation of domestic water demand-supply scenario and its key driving factors in Islamabad-Rawalpindi Metropolitan Area, Pakistan” (ID: PONE-D-23-14511). Those comments are all valuable and very helpful for revising and improving our paper, as well as the important guiding significance to our research. We have studied comments carefully, incorporated all necessary corrections which seemed technically sound, and prepared our arguments for the query as considered pertinent in terms of the difference in opinion on logical grounds. We hope it meets the requirements for your kind approval. All changes in the paper are marked in track change mode in the submitted version of the document. The following are responses to the comments.

Note: Prefix “C” means Comment, Prefix “R” means Response

C-1. Many people around the world suffer from clean water scarcity. The present paper focus on demand and water supply in major connected cities of Pakistan. Though the work done is of comprehensive nature but still more work is required to justify the findings of the study. Urban expansion is compromising the water availability to justify this statement with change in Land use land cover in the area. Image acquisition is not hard when a freeware data set is available on LANDSAT. I would suggest acquiring time series data in order to see the trend and verify the results in terms of land use and land cover change in the area. At least a change over a period of 20 years interval would work and give better results when change in population has been reported.

R-1. LULC results have been incorporated in the manuscript. Needful has been done.

C-2. What data sources were used to estimate domestic water demand in the study?

R-2. Primary data was obtained from concerned civic agencies like Capital Development Authority (CDA), Rawalpindi Water and Sanitation Agency (RWASA) and Rawalpindi Cantonment Broad (RCB) during Feb-July, 2021. Data was collected through extensive interviews, literature/reports provided and visiting their websites during COVID pandemic.

C-3. Are the data reliable and representative of the population in the Islamabad-Rawalpindi Metropolitan Area?

R-3. Primary and secondary data is obtained from authentic sources like CDA, RWASA, RCB, Federal Bureau of Statistics and NIPS. Therefore, data is reliable and representative of the population.

C-4.How was the water demand calculated?

R-4. Data regarding current water demand and supply in Rawalpindi-Islamabad was obtained from concerned civic agencies. Forecasted water demand-supply and their deficit was calculated as described in methodology section of manuscript.

C-5. Did the methodology consider factors such as population growth, household size, and more importantly seasonal variations in water usage?

R-5. Yes the methodology considered demographic factors like population, population growth, urbanization trends in water usage but not included seasonal variations as no reliable data was obtained from concerned civic agencies.

C-6. Were any specific statistical techniques or models used to estimate water demand? If so, can you provide more details about their application and validation?

R-6. We used simple calculations to forecast water demand–supply scenario based in

C-7. What are the identified key driving factors influencing domestic water demand in the Islamabad-Rawalpindi Metropolitan Area?

R-6. Increase in population, migration and urbanization trends along with life style changes are contributed towards increasing water demand in metropolitan region.

C-8. How were these factors determined or selected for analysis?

R-8. Based on extensive literature review demographic factors, housing units and LULU were selected for analysis.

C-9. Were the driving factors analyzed individually or in combination to understand their relative contributions to water demand?

R-9. Individually as well as in combination.

C-10. Can the authors provide insights into the magnitude of each factor's impact?

R-10. This factor is beyond scope of study.

C-11. Were there any unexpected or significant findings regarding the driving factors?

R-11. No.

C-12. Did the study reveal any previously unexplored factors that could influence water demand?

R-12. Yes, climate changes can influence water demand and supply, which is not covered in present study.

C-13. Did the study consider the existing water infrastructure, sources, and their capacity to meet the projected water demand?

R-13.Yes, all three scenario highlighted that water losses can be catered through improvement in existing water infrastructure, billing, water metering, augmented surface water supplies and rainwater harvesting.

C-14. Were there any limitations or challenges identified in the current water supply system or potential leakages in the institutional arrangements that could be improved?

R-14. Climatic changes and seasonal variations were studied in this study.

C-15. Were potential solutions or strategies proposed to address any shortfalls or gaps?

R-15. All probable solutions were proposed in the end of manuscript.

C-16. How does the estimated water demand align with the existing or planned water supply infrastructure in the Twin Cities?

R-16. Yes.

C-17. What are the implications of the study's findings for water resource management in the Islamabad-Rawalpindi Metropolitan Area? How can the results inform policy decisions or water conservation efforts?

R-17. yes.

C-18. Were any specific recommendations provided to policymakers, local authorities, or water management organizations based on the study's outcomes?

R-18. Add points.

C-19. How do the findings of this study contribute to the existing literature on domestic water demand estimation and its driving factors, particularly in the context of urban areas in Pakistan?

R-19. Yes.

C-20. What are the main limitations or potential sources of error in the study's methodology or data that might affect the accuracy or generalizability of the findings?

R-20. This research is designed carefully under supervision of water experts and data is obtained from authentic sources.

C-21. I suggest authors to clearly mention the potential areas for future studies related to domestic water demand estimation or management in urban areas?

R-21.limitations of the study is added in conclusion. Needful has been done.

C-22. The study did not consider climate change as a main driving factor in water scarcity, was there any particular reason for not considering it? If so, please mention it or I strongly suggest authors to add climate change, demographic shifts, seasonal variation in water utilization which could impact water demand in the Islamabad-Rawalpindi Metropolitan Area?

R-22. proposed for future research.

C-23. I strongly advise authors to read the current development made in the policy direction related to the Twin cities where the authorities have passed a parliamentary bill for rainwater utilization in all commercial and residential buildings. This might provide stronger arguments and insights when discussing research results.

R-23. Needful has been done.

Reply – Reviewer #2

Dear Reviewer,

We would like to thank you for your valuable comments about our manuscript titled “Estimation of domestic water demand -supply scenario and its key driving factors in Islamabad-Rawalpindi Metropolitan Area, Pakistan” (ID: PONE-D-23-14511). Those comments are all valuable and very helpful for revising and improving our paper, as well as the important guiding significance to our research. We have studied comments carefully, incorporated all necessary corrections which seemed technically sound, and prepared our arguments for the query as considered pertinent in terms of the difference in opinion on logical grounds. We hope it meets the requirements for your kind approval. All changes in the paper are marked in track change mode in the submitted version of the document. The following are responses to the comments.

Note: Prefix “C” means Comment, Prefix “R” means Response

Reviewer #2: Comments to authors

I had the opportunity to review your paper “Estimation of domestic water demand-supply scenario and its key driving factors in Islamabad-Rawalpindi Metropolitan Area, Pakistan.” The objectives of the study were: 1) to assess current and projected water demand and supply scenario in twin cities; 2) to relate demographic and urban trends as well as the density of tube wells with water table decline in twin cities.

It is very interesting work and has some merits. However, I have some comments and questions that the authors should address them clearly and scientifically.

C1- The introduction- The instruction is too broad. The authors should review and identify clearly (1) the revolution of domestic water demand supply and (2) the key driving factors of high demand and scarcity of domestic water. Also, the introduction would be more interesting if the authors addressed how water demand and scarcity negatively affected the socioeconomic development of a country, esp. Pakistan?

R1. Thank you for the valuable input. Introduction is modified in light of your suggestions.

C2- Methods and Materials- The data collection procedure was not clearly described. For example, L152: The authors explained that the data was acquired from various organizations through extensive interviews. If so, the authors should indicate the number of respondents who were interviewed and who they were? What questions were used to interview? These are important!

R2. We are grateful for your valuable input. We now tried to elaborate Material and methods section to clearly explain the data sources and data analysis process. Questionnaire is added in the Supplementary material (S1 File).

C3- Results: Demographic profile of the study area: What is the main point of this section that the authors want to provide to readers? I think this part should be moved to combine with 2.1.

R3. We appreciate the in depth study of the reviewer. However, demographic profile of the study is the prime factor contributing towards urbanization, urban sprawl and water demand and supply imbalances. This factor is discussed in results in detail to correlate it with LULCC resulting in groundwater decline on one side and increasing water demand on the other side. Placing this information in Material and Method section will create a missing link in result section.

C4: L233-234: …There are approximately 336,182 and 470,588 234 housing units registered in Islamabad and Rawalpindi, respectively… in what year? The authors should add the year to this growth.

R4: This statistical record was obtained from National Housing Census record of 2017. Year is added now. Needful has been done.

C5: These two sections—3.3.1 and 3.3.2. are more like literature/a review of existing documents/reports. Likely, 3.4 is more about reviewing documents/reports

R5: Section 3.3.1 and 3.3.2 display information about situational analysis of groundwater and surface water resources of study area. This information is compiled from CDA, WASA and RCB through questionnaire based interviews. Without this information discussion about supply demand gap will remain incomplete. However, in revised manuscript we tried our level best to remain precise.

C7: I saw only two sentences in L365-366 and L389-390 that the authors likely reported some ideas/findings from the survey? However, it is inconsistent and unclear between what is mentioned in the methodology and these parts. In the methodology, the authors said that they conducted intensive interviews in the methodology, and here in the result, the authors said, “…reported by respondent during field survey.” So what exactly had the author done? Meanwhile, there was no other part of the result section explaining the results/findings from the interview or survey. I doubt it!

R7: Thank you for your very pertinent question. We have added the questionnaire we used for interview based survey in the S1 File and revised material and method section accordingly. Questionnaire based interviews were conducted to gather data regarding water sources, types, distribution and quantities from officials of CDA, WASA and RCB. Data was analyzed keeping in view the information extracted from these interviews.

C8: Conclusion- L480-484: How did the authors come to that conclusion? There was no clue explaining this concern in the result section. Likely in 4.1. Ways toward secure water future. The authors gave many suggestions, but most are from the authors' ideas/perspectives rather than from this research findings. For example, “(i) …an immediate need for formulation of appropriate groundwater 519 legislation/policy….” There is n

---

## [Decision Letter · Decision Letter 1]

7 May 2024

PONE-D-23-14511R1Estimation of domestic water demand-supply scenario and its key driving factors in Islamabad-Rawalpindi Metropolitan Area, Pakistan.PLOS ONE

Dear Dr. Ali,

Thank you for submitting your manuscript to PLOS ONE. After careful consideration, we feel that it has merit but does not fully meet PLOS ONE’s publication criteria as it currently stands. Therefore, we invite you to submit a revised version of the manuscript that addresses the points raised during the review process.

Your manuscript has been evaluated by three reviewers, and their comments are appended below. While Reviewers 1 and 2 are satisfied with your revisions, Reviewer 3 has commented in detail on the methodological detail reported in your study, and has identified additional requirements for further investigation needed to support the conclusions of your study. Please ensure you address each of the reviewers' comments when revising your manuscript.

We look forward to receiving your revised manuscript.

Kind regards,

Hugh Cowley

Staff Editor

PLOS ONE

Additional Editor Comments:

Reviewers' comments:

Reviewer's Responses to Questions

**Comments to the Author**

1. If the authors have adequately addressed your comments raised in a previous round of review and you feel that this manuscript is now acceptable for publication, you may indicate that here to bypass the “Comments to the Author” section, enter your conflict of interest statement in the “Confidential to Editor” section, and submit your "Accept" recommendation.

Reviewer #1: All comments have been addressed

Reviewer #2: All comments have been addressed

Reviewer #3: All comments have been addressed

2. Is the manuscript technically sound, and do the data support the conclusions?

Reviewer #1: Yes

Reviewer #2: Yes

Reviewer #3: Partly

3. Has the statistical analysis been performed appropriately and rigorously? 

Reviewer #1: Yes

Reviewer #2: Yes

Reviewer #3: No

4. Have the authors made all data underlying the findings in their manuscript fully available?

Reviewer #1: Yes

Reviewer #2: Yes

Reviewer #3: No

5. Is the manuscript presented in an intelligible fashion and written in standard English?

Reviewer #1: Yes

Reviewer #2: Yes

Reviewer #3: No

6. Review Comments to the Author

Reviewer #1: The authors have fully addressed all my queries. The issue discussed in article is of utmost significance for a resource lackluster country like Pakistan, and I accept it for publication.

Reviewer #2: Dear Authors,

Thanks for the efforts yo have made to revise the manuscript. I am satisfied with the response and current format.

Good luck!

Reviewer #3: Dear Authors, Please find below a list of recommendations based on the review of your manuscript. I hope these suggestions will be of true value in improving your excellent work.

1. Grammatical, spelling, and style corrections:

Lines 24, 381: Change "MDG" to "MGD" (Million Gallons per Day)

Line 61: Change "was processed" to "were processed"

Lines 64 and 65: Change "protuberate" to "increase". Additionally, please include the corresponding citations and their respective bibliographic references.

Line 137: Change "vary" to "varies"

Line 155: Change "was collected" to "were collected"

Line 174: Change "K1" in the equation (1) to "K_1" (subscript for 1).

Lines 174, 180, Ensure that all equations are numbered and referenced by their number in the paragraph preceding their presentation.

Lines 181 - 182: Please refrain from using the equal sign in the 'Where' section for equations.

Line 200: Change "cataloged" to "categorized"

Line 222: Change "explicated" to "explained"

Line 243: Change "regularized" to "regulated"

Please format citation numbers using square brackets [] instead of parentheses ().

2. Areas where depth of analysis could be improved or clarification is needed:

- The introduction section could benefit from a more thorough review of the existing literature on urban water demand and supply challenges, particularly in the context of rapidly growing cities in developing countries.

The authors should consider incorporating relevant theories and concepts from urban planning, water resource management, and sustainability science to provide a more robust conceptual grounding for their study. This could include discussing theoretical perspectives on urban water demand drivers, urban metabolism, socio-ecological systems, and integrated water resource management approaches.

Additionally, please include the corresponding citations and their respective bibliographic references.

- The methodology section could provide more detailed information on the data collection process, including the specific sources of primary and secondary data, the time period covered, and any limitations or assumptions made in the analysis.

This reviewer suggests that the mathematical model used in the article currently lacks depth. It is recommended to thoroughly enhance this important component of the work.

The authors should consider employing more advanced analytical techniques, such as spatial analysis, statistical modeling, and scenario analysis, to provide a more comprehensive and rigorous assessment of the water demand-supply dynamics in the twin cities. This could include:

a. Incorporating spatial data and Geographic Information Systems (GIS) to analyze the spatial patterns of water demand, supply infrastructure, and demographic factors across the study area.

b. Developing and applying statistical models to quantify the relationships between water demand, urbanization patterns, socioeconomic factors, and climate variability.

c. Employing scenario analysis techniques to explore alternative future scenarios and their potential impacts on water demand and supply, considering various policy interventions, infrastructure investments, and climate change projections.

d. The article primarily focuses on demographic trends and water infrastructure, but could benefit from a more holistic consideration of socio-economic and environmental factors influencing urban water demand and supply. The authors should consider incorporating additional variables, such as household income levels, water pricing mechanisms, industrial and agricultural water use, land-use patterns, and environmental factors like climate change impacts and ecosystem services.

e. To enhance the broader relevance and impact of the study, the authors could consider incorporating a comparative analysis with other rapidly urbanizing cities or regions facing similar water challenges. This could involve benchmarking the twin cities' water demand-supply dynamics against other urban areas, identifying successful practices and lessons learned, and exploring opportunities for cross-city learning and knowledge exchange.

f. Additionally, the authors should describe the validation process for the recorded data, including groundwater level decline and water demand and supply. They should also present the model used for future water demand and supply projection, explain their selection criteria for this model, and detail the conditions under which they validated these projections. This will enhance the credibility and reliability of the study's findings.

- The results section could include a more in-depth analysis of the factors contributing to the water demand-supply gap in the twin cities, such as the impact of rapid urbanization, land-use changes, and climate variability.

- The discussion section could explore the implications of the study findings for sustainable water management in the twin cities and provide more specific recommendations for policy interventions and management strategies.

After the discussion, please include the limitations of the study and suggestions for future work:

- A subsection titled "Limitations" where authors critically discuss the constraints and potential shortcomings of the study. For example, they could acknowledge limitations related to data availability, methodological assumptions, or the generalizability of the findings to other urban contexts.

- Authors should insert a subsection titled "Future Work" where they outline recommendations and directions for future research.

3. Suggestions for improving structure, organization, and coherence:

- The methodology section could be divided into subsections for clarity, such as "Study Area", "Data Collection", "Data Analysis", and "Forecasting Approach".

This reviewer considers it important for the authors to include a flowchart illustrating their research methodology. Such visualization will significantly enhance the clarity and understanding of the study's procedures

- The discussion section could be expanded to include a broader interpretation of the study findings within the larger context of urban water management challenges and potential solutions.

4. This reviewer suggests that authors include a section titled 'Results Validation' or 'Verification of Findings' in their manuscript, within relevant sections such as Methodology, Results, and Discussion. In this section, they should describe the specific methods used to validate or corroborate the results presented in their study. Some strategies they might consider include:

- Triangulation of data sources: Authors should explain how they compared their findings with other independent data sources to ensure accuracy and validity.

- Sensitivity analysis or uncertainty quantification: If applicable, authors should explain how they assessed the robustness of their results to different scenarios or variations in data.

- Comparison with benchmarks or established guidelines: Authors should evaluate how their findings align with accepted standards or guidelines in the field, including software models for calibration, if applicable.

- Statistical validation: Authors should describe any statistical analysis conducted to support the significance of the results. Additionally, authors may consider including an expert review or stakeholder consultation. Authors should detail if they sought feedback or validation from subject matter experts or relevant stakeholders.

7. PLOS authors have the option to publish the peer review history of their article (what does this mean? ). If published, this will include your full peer review and any attached files.

**Do you want your identity to be public for this peer review?** For information about this choice, including consent withdrawal, please see our Privacy Policy .

Reviewer #1: **Yes: ** Wahid Ullah

Reviewer #2: No

Reviewer #3: **Yes: ** Holger Benavides-Muñoz

---

## [Author Response · Author response to Decision Letter 2]

30 May 2024

Reply – Reviewer #3

Dear Reviewer,

We would like to thank you for your valuable comments about our manuscript titled “Estimation of domestic water demand -supply scenario and its key driving factors in Islamabad-Rawalpindi Metropolitan Area, Pakistan” (ID: PONE-D-23-14511). Those comments are all valuable and very helpful for revising and improving our paper, as well as the important guiding significance to our research. We have studied comments carefully, incorporated all necessary corrections which seemed technically sound, and prepared our arguments for the query as considered pertinent in terms of the difference in opinion on logical grounds. We hope it meets the requirements for your kind approval. All changes in the paper are marked in track change mode in the submitted version of the document. The following are responses to the comments.

Note: Prefix “C” means Comment, Prefix “R” means Response

Reviewer #2: Comments to authors

Reviewer #3: Dear Authors, Please find below a list of recommendations based on the review of your manuscript. I hope these suggestions will be of true value in improving your excellent work.

C-1- Grammatical, spelling, and style corrections:

Lines 24, 381: Change "MDG" to "MGD" (Million Gallons per Day)

Line 61: Change "was processed" to "were processed"

Lines 64 and 65: Change "protuberate" to "increase". Additionally, please include the corresponding citations and their respective bibliographic references.

Line 137: Change "vary" to "varies"

Line 155: Change "was collected" to "were collected"

Line 174: Change "K1" in the equation (1) to "K_1" (subscript for 1).

Lines 174, 180, Ensure that all equations are numbered and referenced by their number in the paragraph preceding their presentation.

Lines 181 - 182: Please refrain from using the equal sign in the 'Where' section for equations.

Line 200: Change "cataloged" to "categorized"

Line 222: Change "explicated" to "explained"

Line 243: Change "regularized" to "regulated"

Please format citation numbers using square brackets [] instead of parentheses ().

R-1. Thank you for the valuable input. Grammatical, spelling, and style corrections are modified in light of your suggestions. Needful has been done.

C-2. Areas where depth of analysis could be improved or clarification is needed:

R-2. Introduction Section: Thank you for the valuable input. Introduction is modified in light of your suggestions. Situational analysis of other rapidly growing cities of the world is incorporated (Line No 124-130). Recent literature is incorporated and references were added to improve manuscript.

Methodology Section: Primary data was obtained from concerned civic agencies like Capital Development Authority (CDA), Rawalpindi Water and Sanitation Agency (RWASA) and Rawalpindi Cantonment Broad (RCB) during 2020-2021. Data was collected through extensive interviews, literature/reports provided and visiting their websites during COVID pandemic Primary and secondary data is obtained from authentic sources like CDA, RWASA, RCB, Federal Bureau of Statistics and NIPS. All datasheets were cross verified from reports and available literature. Therefore, data is reliable and representative of the population. We used simple calculations to forecast water demand–supply scenario and statistical and economic modeling is proposed for future studies. We now tried to elaborate Material and methods section to clearly explain the data sources and data analysis process. Questionnaire is added in the Supplementary material (S1 File). Data analysis sheets are added in the Supplementary material (S7 File). Limitations of the study were also incorporated. Methodology flowchart is improved. GIS and Remotely sensed data to investigate LULC have been incorporated in the manuscript. The methodology considered demographic factors like population, population growth, urbanization trends in water usage but not included seasonal variations and climatic changes as no reliable data were obtained from concerned civic agencies. Needful has been done. We are grateful for your valuable input.

Results section:

A. We appreciate the in depth study of the reviewer. Increase in population, migration and urbanization trends along with life style changes are contributed towards increasing water demand in metropolitan region. However, demographic profile of the study is the prime factor contributing towards urbanization, urban sprawl and water demand and supply imbalances. This factor is discussed in results in detail to correlate it with LULCC resulting in groundwater decline on one side and increasing water demand on the other side.

B. All probable solutions and specific recommendations were proposed in the end of manuscript.

Discussion Section:

A. Limitations of the study: limitations of the study is added in the end of discussion section and conclusion. Needful has been done.

B. Future work: future prospects were added in the end of discussion section. Needful has been done.

C-3. Suggestions for improving structure, organization, and coherence:

R-3

A. The methodology section is revised and organized in the light of your comments to improve manuscript quality. Needful has been done.

B. Methodology flowchart is included in the manuscript as Figure No. 2 for more clarity. Needful has been done.

C. Results are now discussed in detail and solutions are proposed in the way forward section of the manuscript.

R-4. This reviewer suggests that authors include a section titled 'Results Validation' or 'Verification of Findings' in their manuscript, within relevant sections such as Methodology, Results, and Discussion. In this section, they should describe the specific methods used to validate or corroborate the results presented in their study. Some strategies they might consider include:

'Verification of Findings' As data is obtained from concerned departments and analyzed under supervision of water experts so results obtained were reliable and authentic.

Accuracy Analysis is included in methodology to validate LULC data collection, analysis and interpretation.

Comparison with benchmark or established guidelines.

R-4 Results were discussed with available literature. LULC results were compared with other studies in the world and Pakistan. Scenarios were built carefully through extensive literature surveys carried out in different part of the world for water demand-supply forecasting.

Statistical validation: Authors should describe any statistical analysis conducted to support the significance of the results. Additionally, authors may consider including an expert review or stakeholder consultation. Authors should detail if they sought feedback or validation from subject matter experts or relevant stakeholders.

R-4. Ambit of the study was to forecast water demand–supply scenario with respect to population projections based on three management options in accordance with environmental conditions of study area under supervision of water experts from PCRWR, CDA, RDA, R-WASA and RCB. All three scenario highlighted that water losses can be catered through improvement in existing water infrastructure, billing, water metering, augmented surface water supplies and rainwater harvesting. This research is designed on request of concerned civic agencies to chairman PCRWR to cater demand-Supply imbalances in twin cities.

Therefore, statistical validation is beyond scope of the study. Present study is focused on the analysis of the existing water resources use and supply, water demand-supply imbalances, key driving factors and LULC change using simple equations for population projections and water demand-supply forecast scenarios. For future work, present study suggest prediction of water situation by the statistical or economic model to validate results of present investigation.

---

## [Decision Letter · Decision Letter 2]

20 Aug 2024

PONE-D-23-14511R2Estimation of domestic water demand-supply scenario and its key driving factors in Islamabad-Rawalpindi Metropolitan Area, Pakistan.PLOS ONE

Dear Dr. Ali,

Thank you for submitting your manuscript to PLOS ONE. After careful consideration, we feel that it has merit but does not fully meet PLOS ONE’s publication criteria as it currently stands. Therefore, we invite you to submit a revised version of the manuscript that addresses the points raised during the review process. Reviewer 4The manuscript has many grammatical issues that need to be corrected. In many places, the scientific explanations are ambiguous and most of the sentences are not clear. The authors need to proofread the manuscript thoroughly. I suggest the following changes and improvements:1. The authors need to improve the Abstract section considering the important findings and underscore the scientific value added to your manuscript in your abstract.2. Keywords are a beneficial tool for search engines to find relevant material. The author needs to improve, be concise, and avoid abbreviations.3. The current structure of the introduction is not well written and is too long. Please revise and improve it. Additionally, the last part of the introduction needs to be further improved considering the main theme/objectives and findings of the research.4. The number of references in the introduction section needs to be extended to support your statement/data. 5. Pakistan is the one of the most populous country in the world with a total………… Please improve the sentence and eliminate the grammatical issue.6. What are the current research gaps and significance of this work? Please mention this clearly in the introduction section.7. For the purpose, the demographic trends and water demand and supply forecasting scenario………. Improve the sentence.8. All Figures and Tables legends/titles should provide enough information.9. The author needs to care about prepositions especially “the” in the text throughout the manuscript. 10. For the consistency of the manuscript, the words should be consistent throughout the text according to the journal, such as (sq km or km2), (table 5 or Table 5), etc.11. Conclusion is a fundamental recap of whole data sets. The authors need to improve based on key findings with clearer summarized information and research significance. 12. All references should be checked as per the Journal format.

We look forward to receiving your revised manuscript.

Kind regards,

Asghar Khan

Academic Editor

PLOS ONE

Reviewers' comments:

Reviewer's Responses to Questions

**Comments to the Author**

1. If the authors have adequately addressed your comments raised in a previous round of review and you feel that this manuscript is now acceptable for publication, you may indicate that here to bypass the “Comments to the Author” section, enter your conflict of interest statement in the “Confidential to Editor” section, and submit your "Accept" recommendation.

Reviewer #3: All comments have been addressed

Reviewer #4: (No Response)

2. Is the manuscript technically sound, and do the data support the conclusions?

Reviewer #3: Yes

Reviewer #4: Partly

3. Has the statistical analysis been performed appropriately and rigorously? 

Reviewer #3: Yes

Reviewer #4: N/A

4. Have the authors made all data underlying the findings in their manuscript fully available?

Reviewer #3: Yes

Reviewer #4: (No Response)

5. Is the manuscript presented in an intelligible fashion and written in standard English?

Reviewer #3: Yes

Reviewer #4: No

6. Review Comments to the Author

Reviewer #3: I have carefully reviewed the requested changes to your manuscript and can confirm that you have fully complied with all the established modifications. The adjustments made are of good quality and relevant.

I appreciate the effort and dedication invested in addressing the suggestions and correcting the indicated aspects.

Reviewer #4: Comments:

The manuscript has many grammatical issues that need to be corrected. In many places, the scientific explanations are ambiguous and most of the sentences are not clear. The authors need to proofread the manuscript thoroughly. I suggest the following changes and improvements:

1. The authors need to improve the Abstract section considering the important findings and underscore the scientific value added to your manuscript in your abstract.

2. Keywords are a beneficial tool for search engines to find relevant material. The author needs to improve, be concise, and avoid abbreviations.

3. The current structure of the introduction is not well written and is too long. Please revise and improve it. Additionally, the last part of the introduction needs to be further improved considering the main theme/objectives and findings of the research.

4. The number of references in the introduction section needs to be extended to support your statement/data.

5. Pakistan is the one of the most populous country in the world with a total………… Please improve the sentence and eliminate the grammatical issue.

6. What are the current research gaps and significance of this work? Please mention this clearly in the introduction section.

7. For the purpose, the demographic trends and water demand and supply forecasting scenario………. Improve the sentence.

8. All Figures and Tables legends/titles should provide enough information.

9. The author needs to care about prepositions especially “the” in the text throughout the manuscript.

10. For the consistency of the manuscript, the words should be consistent throughout the text according to the journal, such as (sq km or km2), (table 5 or Table 5), etc.

11. Conclusion is a fundamental recap of whole data sets. The authors need to improve based on key findings with clearer summarized information and research significance.

12. All references should be checked as per the Journal format.

7. PLOS authors have the option to publish the peer review history of their article (what does this mean? ). If published, this will include your full peer review and any attached files.

**Do you want your identity to be public for this peer review?** For information about this choice, including consent withdrawal, please see our Privacy Policy .

Reviewer #3: **Yes: ** Benavides-Muñoz Holger M.

Reviewer #4: No

---

## [Author Response · Author response to Decision Letter 3]

17 Oct 2024

Estimation of domestic water demand -supply scenario and its key driving factors in Islamabad-Rawalpindi Metropolitan Area, Pakistan.

Manuscript ID: PONE-D-23-14511

Reply – Reviewer #4

Dear Reviewer,

We would like to thank you for your valuable comments about our manuscript titled “Estimation of domestic water demand-supply scenario and its key driving factors in Islamabad-Rawalpindi Metropolitan Area, Pakistan” (ID: PONE-D-23-14511). Those comments are all valuable and very helpful for revising and improving our paper, as well as the important guiding significance to our research. We have studied comments carefully, incorporated all necessary corrections which seemed technically sound, and prepared our arguments for the query as considered pertinent in terms of the difference in opinion on logical grounds. We hope it meets the requirements for your kind approval. All changes in the paper are marked in track change mode in the submitted version of the document. The following are responses to the comments.

Note: Prefix “C” means Comment, Prefix “R” means Response

C-1. The authors need to improve the Abstract section considering the important findings and underscore the scientific value added to your manuscript in your abstract.

R-1. Changes have been incorporated in the Abstract section of manuscript. Needful has been done.

C-2. Keywords are a beneficial tool for search engines to find relevant material. The author needs to improve, be concise, and avoid abbreviations.

R-2. Keywords have been improved in the manuscript and abbreviations were also removed. Needful has been done.

C-3. The current structure of the introduction is not well written and is too long. Please revise and improve it. Additionally, the last part of the introduction needs to be further improved considering the main theme/objectives and findings of the research.

R-3. Introduction has been revised and improved in the light of comments. Needful has been done.

C-4. The number of references in the introduction section needs to be extended to support your statement/data.

R-4. References used in the introduction has been revised, supported, correlate with current study and improved in the light of comments. Needful has been done.

C-5. Pakistan is the one of the most populous country in the world with a total………… Please improve the sentence and eliminate the grammatical issue.

R-5. Grammatical errors were fixed throughout the manuscript. Needful has been done.

C-6. What are the current research gaps and significance of this work? Please mention this clearly in the introduction section.

R-6. The current research gaps and significance of this work was revised in the last paragragh of introduction. Needful has been done

C-7. For the purpose, the demographic trends and water demand and supply forecasting scenario………. Improve the sentence.

R-7. Grammatical errors were fixed throughout the manuscript. Needful has been done.

C-8. All Figures and Tables legends/titles should provide enough information.

R-8. All Figures and Tables legends/titles have been revised. Needful has been done.

C-9. The author needs to care about prepositions especially “the” in the text throughout the manuscript.

R-9. Prepositions have been fixed throughout the manuscript. Needful has been done.

C-10. For the consistency of the manuscript, the words should be consistent throughout the text according to the journal, such as (sq km or km2), (table 5 or Table 5), etc.

R-10. Words have been fixed throughout the manuscript. Needful has been done.

C-11. Conclusion is a fundamental recap of whole data sets. The authors need to improve based on key findings with clearer summarized information and research significance.

R-11. Conclusion has been revised in the manuscript. Needful has been done.

C-12. All references should be checked as per the Journal format.

R-12. All references have been rechecked as per the Journal format. Needful has been done.

---

## [Decision Letter · Decision Letter 3]

22 Nov 2024

PONE-D-23-14511R3Estimation of the domestic water demand-supply scenario and its key driving factors in the Islamabad-Rawalpindi Metropolitan Area, Pakistan.PLOS ONE

Dear Dr. Ali,

Thank you for submitting your manuscript to PLOS ONE. After careful consideration, we feel that it has merit but does not fully meet PLOS ONE’s publication criteria as it currently stands. Therefore, we invite you to submit a revised version of the manuscript that addresses the points raised during the review process.

**Manuscript Title:**
*Estimation of the Domestic Water Demand-Supply Scenario and Its Key Driving Factors in the Islamabad-Rawalpindi Metropolitan Area, Pakistan*

Dear Dr. Maria,

Thank you for submitting your manuscript for consideration in PLOS ONE. After peer review, the following concerns were raised, which require your attention to enhance the rigor and applicability of the study:

<h3>**Reviewer Comments** </h3>

**Statistical Analysis:**The reviewer requested details about any statistical analysis conducted to validate the significance of your results. Statistical tools are typically essential for ensuring reliability and robustness, particularly in studies involving demand-supply forecasting and resource imbalances.**Stakeholder Consultation:**The reviewer suggested including feedback from subject matter experts or stakeholders to validate your findings and improve the practical relevance of the study. Stakeholder engagement is especially valuable in applied research to align results with real-world needs and scenarios.

<h3>**Authors' Response** </h3>

In your response, you stated that:

Statistical validation is beyond the scope of the study, which focuses on analyzing water resources, demand-supply imbalances, and land use/land cover (LULC) changes using simple equations.Future work may incorporate statistical or economic models for validation.

<h3>**Assessment of Your Response** </h3>

While we understand your perspective, the response raises the following concerns:

**Lack of Statistical Validation:**The absence of statistical analysis weakens the reliability of your conclusions. Even basic statistical methods could enhance the credibility of your projections and findings.Simply stating that statistical analysis is beyond the study’s scope does not adequately address the reviewer’s concern.
**Absence of Stakeholder Feedback:**The lack of input from stakeholders or subject matter experts diminishes the practical relevance of your findings. Applied research in resource management benefits significantly from stakeholder insights, which could validate and refine assumptions.
**Future Work Suggestion:**While suggestions for future work are appreciated, they do not address the methodological gaps in the current study. Statistical and stakeholder-based validations are fundamental to the present research objectives.

<h3>**Required Revisions** </h3>

To address these concerns, please consider the following:

**Statistical Analysis:**Include at least a minimal statistical analysis to support your results, or provide a strong, detailed rationale for why statistical tools were not used. If computational limitations existed, this should be transparently discussed.
**Stakeholder Consultation:**Provide a justification for the absence of stakeholder engagement, or retrospectively include feedback from relevant experts or stakeholders to strengthen your conclusions.
**Transparent Discussion:**Acknowledge any methodological limitations in the discussion section of the manuscript, and clearly outline how these might affect the interpretation and generalizability of your findings.

<h3>**Next Steps** </h3>

We encourage you to carefully address these points and submit a revised version of your manuscript, along with a point-by-point response to the reviewer’s and editor’s comments. If you believe certain revisions are not feasible, please provide a detailed explanation in your response.

We look forward to receiving your revised manuscript.

Kind regards,

Asghar Khan

Academic Editor

PLOS ONE

Journal Requirements:

Additional Editor Comments :

Manuscript Title: Estimation of the Domestic Water Demand-Supply Scenario and Its Key Driving Factors in the Islamabad-Rawalpindi Metropolitan Area, Pakistan

Dear Dr. Syeda Maria Ali,

Thank you for submitting your manuscript for consideration in PLOS ONE. After peer review, the following concerns were raised, which require your attention to enhance the rigor and applicability of the study:

Reviewer Comments

Statistical Analysis:

The reviewer requested details about any statistical analysis conducted to validate the significance of your results. Statistical tools are typically essential for ensuring reliability and robustness, particularly in studies involving demand-supply forecasting and resource imbalances.

Stakeholder Consultation:

The reviewer suggested including feedback from subject matter experts or stakeholders to validate your findings and improve the practical relevance of the study. Stakeholder engagement is especially valuable in applied research to align results with real-world needs and scenarios.

Authors' Response

In your response, you stated that:

Statistical validation is beyond the scope of the study, which focuses on analyzing water resources, demand-supply imbalances, and land use/land cover (LULC) changes using simple equations.

Future work may incorporate statistical or economic models for validation.

Assessment of Your Response

While we understand your perspective, the response raises the following concerns:

Lack of Statistical Validation:

The absence of statistical analysis weakens the reliability of your conclusions. Even basic statistical methods could enhance the credibility of your projections and findings.

Simply stating that statistical analysis is beyond the study’s scope does not adequately address the reviewer’s concern.

Absence of Stakeholder Feedback:

The lack of input from stakeholders or subject matter experts diminishes the practical relevance of your findings. Applied research in resource management benefits significantly from stakeholder insights, which could validate and refine assumptions.

Future Work Suggestion:

While suggestions for future work are appreciated, they do not address the methodological gaps in the current study. Statistical and stakeholder-based validations are fundamental to the present research objectives.

Required Revisions

To address these concerns, please consider the following:

Statistical Analysis:

Include at least a minimal statistical analysis to support your results, or provide a strong, detailed rationale for why statistical tools were not used. If computational limitations existed, this should be transparently discussed.

Stakeholder Consultation:

Provide a justification for the absence of stakeholder engagement, or retrospectively include feedback from relevant experts or stakeholders to strengthen your conclusions.

Transparent Discussion:

Acknowledge any methodological limitations in the discussion section of the manuscript, and clearly outline how these might affect the interpretation and generalizability of your findings.

Next Steps

We encourage you to carefully address these points and submit a revised version of your manuscript, along with a point-by-point response to the reviewer’s and editor’s comments. If you believe certain revisions are not feasible, please provide a detailed explanation in your response.

We look forward to your revised submission.

Sincerely,

Asghar

Reviewers' comments:

Reviewer's Responses to Questions

**Comments to the Author**

1. If the authors have adequately addressed your comments raised in a previous round of review and you feel that this manuscript is now acceptable for publication, you may indicate that here to bypass the “Comments to the Author” section, enter your conflict of interest statement in the “Confidential to Editor” section, and submit your "Accept" recommendation.

Reviewer #3: All comments have been addressed

Reviewer #4: (No Response)

2. Is the manuscript technically sound, and do the data support the conclusions?

Reviewer #3: Yes

Reviewer #4: (No Response)

3. Has the statistical analysis been performed appropriately and rigorously? 

Reviewer #3: No

Reviewer #4: (No Response)

4. Have the authors made all data underlying the findings in their manuscript fully available?

Reviewer #3: Yes

Reviewer #4: (No Response)

5. Is the manuscript presented in an intelligible fashion and written in standard English?

Reviewer #3: Yes

Reviewer #4: (No Response)

6. Review Comments to the Author

Reviewer #3: The reviewer requested the following:

"Authors should describe any statistical analysis conducted to support the significance of the results. Additionally, authors may consider including an expert review or stakeholder consultation. Authors should detail if they sought feedback or validation from subject matter experts or relevant stakeholders."

However, the authors responded with the following text, which is highly debatable and not necessarily justifiable:

 ..."statistical validation is beyond scope of the study. Present study is focused on the

analysis of the existing water resources use and supply, water demand-supply imbalances, key

driving factors and LULC change using simple equations for population projections and water

demand-supply forecast scenarios. For future work, present study suggest prediction of water

situation by the statistical or economic model to validate results of present investigation."

Reviewer #4: The author has addressed all the comments satisfactorily. However, I would recommend manuscript for publication in this form.

7. PLOS authors have the option to publish the peer review history of their article (what does this mean? ). If published, this will include your full peer review and any attached files.

**Do you want your identity to be public for this peer review?** For information about this choice, including consent withdrawal, please see our Privacy Policy .

Reviewer #3: No

Reviewer #4: No

---

## [Author Response · Author response to Decision Letter 4]

3 Jan 2025

Estimation of the domestic water demand‒supply scenario and its key driving factors in the Islamabad-Rawalpindi Metropolitan Area, Pakistan.

Manuscript ID: PONE-D-23-14511

Reply – Reviewer #5

Dear Reviewer,

We would like to thank you for your valuable comments about our manuscript titled “Estimation of the domestic water demand‒supply scenario and its key driving factors in the Islamabad-Rawalpindi Metropolitan Area, Pakistan.” (ID: PONE-D-23-14511). Those comments are all valuable and very helpful for revising and improving our paper, as well as the important guiding significance to our research. We have studied comments carefully, incorporated all necessary corrections which seemed technically sound, and prepared our arguments for the query as considered pertinent in terms of the difference in opinion on logical grounds. We hope it meets the requirements for your kind approval. All changes in the paper are marked in track change mode in the submitted version of the document. The following are responses to the comments.

Note: Prefix “C” means Comment, Prefix “R” means Response

C-1. Statistical Analysis

R-1. Error bars and trend lines have been incorporated in all figures. Needful has been done.

C-2. Stakeholder Consultation

R-2. Thank you so much for suggestion to improve quality of the manuscript. Stakeholder consultation part is incorporated in the methodology section of the manuscript. Detail description of all stakeholders is given in supplementary material S-8.

C-3. All references should be checked as per the Journal format.

R-3. All references have been rechecked as per the Journal format. Needful has been done.

C-4. Transparent discussion

R-4. Following stakeholders were already consulted and incorporating their suggestions and expert opinions while designing, executing and implementing present study. Needful has been done.

---

## [Editor Report · Decision Letter 4]

14 Jan 2025

Estimation of the domestic water demand-supply scenario and its key driving factors in the Islamabad-Rawalpindi Metropolitan Area, Pakistan.

PONE-D-23-14511R4

Dear Dr. Syeda Maria Ali,

We’re pleased to inform you that your manuscript has been judged scientifically suitable for publication and will be formally accepted for publication once it meets all outstanding technical requirements.

Kind regards,

Asghar Khan

Academic Editor

PLOS ONE
---

## [Editor Report · Acceptance letter]

PONE-D-23-14511R1

Estimation of domestic water demand-supply scenario and its key driving factors in Islamabad-Rawalpindi Metropolitan Area, Pakistan.

Dear Dr. Ali:

I'm pleased to inform you that your manuscript has been deemed suitable for publication in PLOS ONE. Congratulations! Your manuscript is now with our production department.

Kind regards,

on behalf of

Dr. Muhammad Tayyab Sohail

Academic Editor

PLOS ONE